# MODEL-AUGMENTED PRIORITIZED EXPERIENCE REPLAY

**Youngmin Oh**[1], **Jinwoo Shin**[2], **Eunho Yang**[2,3], **Sung Ju Hwang**[2,3]
[1] Samsung Advanced Institute of Technology
[2] Korea Advanced Institute of Science and Technology
[3] AITRICS
youngmin0.oh@samsung.com
{jinwoos, eunhoy, sjhwang82}@kaist.ac.kr

## ABSTRACT

Experience replay is an essential component in off-policy model-free reinforcement learning (MfRL). Due to its effectiveness, various methods for calculating priority scores on experiences have been proposed for sampling. Since critic networks are crucial to policy learning, TD-error, directly correlated to $Q$-values, is one of the most frequently used features to compute the scores. However, critic networks often under- or overestimate $Q$-values, so it is often ineffective to learn for predicting $Q$-values by sampled experiences based heavily on TD-error. Accordingly, it is valuable to find auxiliary features, which positively support TD-error in calculating the scores for efficient sampling. Motivated by this, we propose a novel experience replay method, which we call model-augmented prioritized experience replay (MaPER), that employs new learnable features driven from components in model-based RL (MbRL) to calculate the scores on experiences. The proposed MaPER brings the effect of curriculum learning for predicting $Q$-values better by the critic network with negligible memory and computational overhead compared to the vanilla PER. Indeed, our experimental results on various tasks demonstrate that MaPER can significantly improve the performance of the state-of-the-art off-policy MfRL and MbRL which includes off-policy MfRL algorithms in its policy optimization procedure.

## 1 INTRODUCTION

Experience replay (Lin, 1992; Mnih et al., 2015), which provides experiences that different policies may collect, is an essential component of policy training in reinforcement learning (RL). By utilizing many past experiences in a large buffer, experience replay can stabilize and improve policy training. As a result, experience replay with the buffer has significantly contributed to the success of RL on various tasks (Fujimoto et al., 2018; Haarnoja et al., 2018a;b; Mnih et al., 2015).

Due to its importance, various methods for calculating priority scores of experiences in the buffer have been proposed to sample experiences efficiently (Schaul et al., 2016; Zha et al., 2019; Sinha et al., 2020). Since well-trained critic networks lead to effective policy learning, one of the most popular methods is to utilize pre-defined metrics for prioritizing experiences based on the temporal difference error (TD-error), directly related to the loss of critic networks (Schaul et al., 2016). The prioritized experience replay based on TD-error (PER) has indeed proved its effectiveness in $Q$-learning (Hessel et al., 2018; Schaul et al., 2016).

However, measuring $Q$-values requires to predict the expectation of returns, which can be obtained after multi-steps, so learning to predict $Q$-values generally needs a lot of interactions with an environment. Due to the difficulty of the multi-step estimation, sampling based only on high TD-errors may be often ineffective or even degrade the sample-efficiency of the RL framework on some tasks, compared to the uniform sampling, e.g., see Figure 6 in (Hessel et al., 2018). Moreover, Zha et al. (2019); Wang & Ross (2019); Sinha et al. (2020) show that the effectiveness of PER is questionable under some settings other than $Q$-learning, e.g., policy-based methods. In particular, Zha et al. (2019) reported that transitions with low TD-errors could be suitable for training a policy for some tasks. To overcome this issue, several works attempt to design learning-based prioritizing

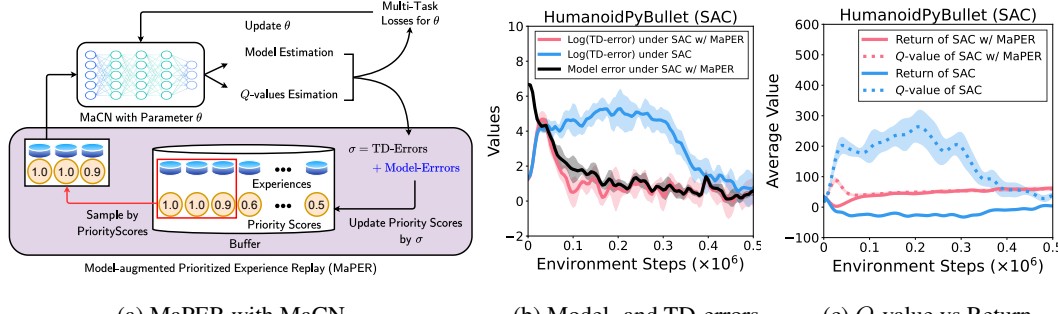

(a) MaPER with MaCN          (b) Model- and TD-errors          (c) $Q$-value vs Return

Figure 1: (a) A high-level illustration of Model-augmented Prioritized Experience Replay (MaPER) with Model-augmented Critic Network (MaCN). MaCN obtains more accurate estimation of $Q$-values via MaPER across the $Q$-value estimator and the model estimator. (b) Curves of TD-errors and model errors from sampled experiences under SAC (Haarnoja et al., 2018a). MaPER leads to a much faster decay of TD-errors. (c) Curves of $Q$-value estimations and returns evaluated by a policy of SAC. MaPER leads to a much faster convergence of $Q$-values to returns. HumanoidPyBullet is a task belonging to Pybullet Gymporium (Ellenberger, 2018–2019). The solid lines and shaded regions denote the mean and standard deviations over the five random runs.

strategies (Zha et al., 2019; Sinha et al., 2020; Oh et al., 2021) utilizing various features, including transition's elements and quantities driven by critic networks. Here, our motivating observation is that all learnable features used in the existing schemes are from critic networks. For instance, although Zha et al. (2019) utilizes multiple features, e.g., reward, TD-error, and timestep, TD-error is only learnable and is obtained from critic networks.

In this paper, we aim for developing a new prioritizing strategy by employing learnable auxiliary features outside critic networks, but can support TD-errors for efficient critics training. To this end, we consider components in Model-based RL (MbRL). To be specific, we first modify the critic network by additionally predicting the reward and the transition with shared weights, which we call *Model-augmented Critic Network* (MaCN). Then, we propose a modified experience replay scheme, coined Model-augmented Prioritized Experience Replay (MaPER), which prioritizes the past experiences having both high model estimation errors and the TD-errors (see Figure 1.(a)). Namely, MaPER encourages MaCN to predict both the $Q$-value and the model (i.e., reward and transition) well. Here, our intuition is two-fold:

- We expect that prioritizing samples additionally with high model estimation errors helps decrease the model errors at the early stage of training, even if MaCN's estimation of the $Q$-value is poor.

- Once MaCN has small model errors, MaPER prefers samples of high TD-error. Here, the learned representation to predict the model well is also useful to predict $Q$-value, as $Q$-value is decided by the model (Sutton & Barto, 2018).

One may suggest to train MaCN without using MaPER, but this is not effective (see Figure 4.(a)). This is because MaPER brings a type of curriculum learning effect (Bengio et al., 2009) to learn $Q$-value well by MaCN; it gradually increases the complexity of samples for training, from high model estimation errors to high TD-errors. Indeed, Figure 1.(b) shows that TD-errors start to abruptly reduce (around $0.05 \times 10^6$ steps) after model errors sufficiently reduce. Figure 1.(b)-(c) also shows that MaPER improves both the decay rate of TD-errors and the convergence rate of $Q$-values to returns.

To summarize, our contributions are as follows:

- We propose *Model-augmented Prioritized Experience Replay* (MaPER), which computes the priority scores of the past experiences additionally considering the model estimation error. MaPER helps reduce TD-errors much faster than learning the $Q$-values alone, as shown in Figure 1.(b)-(c). It results in large improvement in the return with the comparable number of parameters of the original critic networks.

- MaPER can be seamlessly integrated into any modern off-policy RL frameworks with critic networks, including both MfRL (e.g. SAC (Haarnoja et al., 2018a), TD3 (Fujimoto et al., 2018)

and Rainbow (van Hasselt et al., 2019)) and MbRL (e.g. MBPO (Janner et al., 2019)) methods with negligible memory or computational overhead.

We remark that to the best of our knowledge, none of the existing methods improve RL's experience replay by employing model learning although there exist some relevant works for different purposes, e.g., to augment rewards for improving the exploration (Pathak et al., 2017; Shelhamer et al., 2016) or to enhance representation learning for pixel-based observations (Lin et al., 2019; Agarwal et al., 2021; Stooke et al., 2021; Lee et al., 2020b; Schrittwieser et al., 2020; Hessel et al., 2021; Hafner et al., 2020b; Srinivas et al., 2020; Guo et al., 2020; Lee et al., 2019).

## 2 METHOD

### 2.1 PRELIMINARY

In an RL framework, an agent interacts with an environment which consists of reward and transition distributions, $\mathcal{R} : S \times A \to \Delta(\mathbb{R})$ and $\mathcal{T} : S \times A \to \Delta(S)$, for state and action spaces $S$ and $A$, where $\Delta(\cdot)$ denotes a set of distributions. The agent selects an action $a_t \sim \pi(\cdot|s_t)$ by its policy $\pi$ on the current state $s_t \in S$ to receive the next state $s_{t+1} \sim \mathcal{T}(s_t, a_t)$ and a reward $r_t \sim \mathcal{R}(s_t, a_t)$ over discrete timesteps $t$. The agent's objective is to learn a policy $\pi$ that maximizes $\mathbb{E}_{s_0 \sim \rho, \pi} \left[ \sum_{k=0}^{\infty} \gamma^k r_k \right]$, where $\rho$ is a distribution of initial state $s_0$ at each episode and $\sum_{k=0}^{\infty} \gamma^k r_{t+k}$ is the discounted cumulative rewards with a discount factor $\gamma \in [0, 1)$.

Although our method is generally applicable to off-policy RL methods which have $Q$-networks, i.e., critics, throughout this section, we focus on off-policy actor-critic RL algorithms which consist of the policy (i.e., actor) $\pi_\Theta(a|s)$ and critic networks $Q_\theta(s, a)$ with an experience replay buffer $\mathcal{B}$, where $\Theta$ and $\theta$ are their parameters, respectively. The most commonly used loss for $Q_\theta(s, a)$ (Mnih et al., 2015; Haarnoja et al., 2018a; Fujimoto et al., 2018) is:

$$\mathbb{E}_{(s_t, a_t, r_t, s_{t+1}) \sim \mathcal{B}} \left[ \| \delta_t Q_\theta^{\pi_\Theta} \|_{\mathrm{MSE}} \right], \tag{1}$$

where $\| \cdot \|_{\mathrm{MSE}}$ is the mean-squared error and $\delta_t Q_\theta^{\pi_\Theta}$ is the Temporal Difference error (TD-error) defined as follows: for a given transition $(s_t, a_t, r_t, s_{t+1})$,

$$\delta_t Q_\theta^{\pi_\Theta} = \delta Q_\theta^{\pi_\Theta}(s_t, a_t, r_t, s_{t+1}) = r_t + \gamma \mathbb{E}_{a' \sim \pi_\Theta(\cdot|s_{t+1})} \left[ Q_\theta(s_{t+1}, a') \right] - Q_\theta(s_t, a_t). \tag{2}$$

We can interpret this value as a measure of how surprising or 'unexpected' the transition is.

### 2.2 MODEL-AUGMENTED PRIORITIZED EXPERIENCE REPLAY (MaPER)

**Model-augmented Critic Network (MaCN).** An environment is depicted as reward and transition distributions, $\mathcal{R}_\theta(s, a)$ and $\mathcal{T}_\theta(s, a)$. We observe that these two distributions and $Q_\theta(s, a)$ have the same input domain, i.e., $S \times A$, the product of state and action spaces. Motivated by it, we slightly modify $Q_\theta(s, a)$ into $\mathcal{C}_\theta(s, a)$ to additionally predict the environment by a reward model $\mathcal{R}_\theta(s, a)$ and a transition model $\mathcal{T}_\theta(s, a)$ in parallel via parameter sharing. We refer to $\mathcal{C}_\theta(s, a)$ as a *Model-augmented Critic Network* (MaCN) such that $\mathcal{C}_\theta = (\mathcal{Q}_\theta, \mathcal{R}_\theta, \mathcal{T}_\theta)$. Here, $\mathcal{Q}_\theta$ measures $Q$-values with the reward estimation $\mathcal{R}_\theta$ as (Hessel et al., 2021) to stimulate the exploration (see Table C.1 in the supplementary material). In other words, we formulate the loss for $\mathcal{Q}_\theta$ using the estimated rewards, that is similar to Eq. (1):

$$L_{\mathcal{Q}_\theta} = \mathbb{E}_{(s_t, a_t, s_{t+1}) \sim \mathcal{B}} \left[ \| \delta_t \mathcal{Q}_\theta^{\pi_\Theta}(s_t, a_t, s_{t+1}) \|_{\mathrm{MSE}} \right], \tag{3}$$

where

$$\delta \mathcal{Q}_\theta^{\pi_\Theta}(s_t, a_t, s_{t+1}) = \mathcal{Q}_\theta(s_t, a_t) - \left( \mathcal{R}_\theta(s_t, a_t) + \gamma \mathbb{E}_{a' \sim \pi_\Theta(\cdot, s_{t+1})} \left[ \mathcal{Q}_\theta(s_{t+1}, a') \right] \right). \tag{4}$$

Here, $\mathcal{R}_\theta$ is detached in the backward propagation of the loss $L_{\mathcal{Q}_\theta}$. Then the loss for $\mathcal{C}_\theta$ is

$$L_{\mathcal{C}_\theta} = \xi_1 L_{\mathcal{Q}_\theta} + \xi_2 L_{\mathcal{R}_\theta} + \xi_3 L_{\mathcal{T}_\theta}, \tag{5}$$

where $L_{\mathcal{R}_\theta}$ and $L_{\mathcal{T}_\theta}$ are the losses for $\mathcal{R}_\theta$ and $\mathcal{T}_\theta$, respectively, with adaptively changing positive coefficients $\xi_1$, $\xi_2$, and $\xi_3$ by employing a dynamic method in (Liang & Zhang, 2020; Liu et al., 2019b). Assuming deterministic environments, we employ the following losses for $\mathcal{R}_\theta$ and $\mathcal{T}_\theta$:

$$L_{\mathcal{R}_\theta} = \mathbb{E}_{(s_t, a_t, r_t, s_{t+1}) \sim \mathcal{B}} \left[ \| \delta_t \mathcal{R}_\theta \|_{\mathrm{MSE}} \right], \quad L_{\mathcal{T}_\theta} = \mathbb{E}_{(s_t, a_t, r_t, s_{t+1}) \sim \mathcal{B}} \left[ \| \delta_t \mathcal{T}_\theta \|_{\mathrm{MSE}} \right],$$

---

**Algorithm 1** Model-augmented Prioritized Experience Replay based on Actor-Critic Methods

---

1: Initialize the model-augmented critic network's parameters $\theta$, the actor's parameters $\Theta$, a replay buffer $\mathcal{B} \leftarrow \emptyset$, priority set $\mathcal{P}_\mathcal{B} \leftarrow \emptyset$, and the batch size $m$
2: **for** each timestep $t$ **do**
3:     Choose $a_t$ from the actor and collect a transition $(s_t, a_t, r_t, s_{t+1})$ from the environment
4:     Update replay buffer $\mathcal{B} \leftarrow \mathcal{B} \cup \{(s_t, a_t, r_t, s_{t+1})\}$ and priority set $\mathcal{P}_\mathcal{B} \leftarrow \mathcal{P}_\mathcal{B} \cup \{\max_\mathcal{B} \mathcal{P}_\mathcal{B}\}$
5:     **for** each gradient step **do**
6:         Sample an index $I$ by the given set $\mathcal{P}_\mathcal{B}$ and Eq. (8) with $|I| = m$
7:         Calculate weights $\{w_i\}_{i \in I}$ by Eq. (9)
8:         Learn $\theta$ by Eq. (5) and $\Theta$ by $\{\mathcal{B}_i\}_{i \in I} \subset \mathcal{B}$ with corresponding weights $\{w_i\}_{i \in I}$
9:         Update a priority set $\{\sigma_i\}_{i \in I}$ by Eq. (10)
10:    **end for**
11: **end for**

---

where

$$\delta_t \mathcal{R}_\theta = \mathcal{R}_\theta(s_t, a_t) - r_t, \ \delta_t \mathcal{T}_\theta = \mathcal{T}_\theta(s_t, a_t) - s_{t+1}.$$

**Formulation of MaPER.** We first explain the concept of PER (Schaul et al., 2016) since we will augment it with the model-based components. Let $[n]$ be defined as the set $\{1, \cdots, n\}$ for a positive integer $n$. Without loss of generality, we can suppose that the replay buffer $\mathcal{B}$ stores the following information as its $i$-th transition:

$$\mathcal{B}_i = \left(s_{\kappa(i)}, a_{\kappa(i)}, r_{\kappa(i)}, s_{\kappa(i)+1}\right), \tag{6}$$

with a function $\kappa(i)$ from the index of $\mathcal{B}$ to a corresponding timestep. PER calculates each $\mathcal{B}_i$'s priority $\sigma_i$ as the recently computed TD-error for itself and collects a set of priority scores:

$$\mathcal{P}_\mathcal{B} = \{\sigma_1, \cdots, \sigma_{|\mathcal{B}|}\}, \tag{7}$$

where each priority $\sigma_i$ is updated whenever the corresponding transition is sampled for training the actor and critic networks. The TD-error (Eq. (2)) is the most frequently used quantity to make the priority set (Eq. (7)) (Schaul et al., 2016; Brittain et al., 2019; Hessel et al., 2018; van Hasselt et al., 2019). Then the sampling strategy of PER is to determine an index set $I$ in $[|\mathcal{B}|]$ from the probability $p_i$ of $i$-th transition defined by the priority set:

$$p_i = \frac{\sigma_i^\alpha}{\sum_{k \in [|\mathcal{B}|]} \sigma_k^\alpha}, \tag{8}$$

with a hyper-parameter $\alpha > 0$. Finally, we calculate the importance-sampling weights as follows:

$$w_i = \left(\frac{1}{|\mathcal{B}|p_i}\right)^\beta, \tag{9}$$

where $\beta > 0$ is also a hyper-parameter to compensate the bias of probabilities.

We design a new priority score equation with MaCN. Since our method uses $\mathcal{C}_\theta(s, a)$ instead of the original critic network, we modify the rule in obtaining the priority set in Eq. (7) for experiences in the buffer $\mathcal{B}$ accordingly. To this end, we compute the priority of each transition as the sum of the TD-errors and model errors, which is used to compute probabilities and weights for corresponding transitions by Eqs. (8)-(9), as follows:

$$\sigma_i = \xi_1 \|\delta_{\kappa(i)} \mathcal{Q}_\theta^{\pi_\Theta}\|_{\text{MSE}} + \xi_2 \|\delta_{\kappa(i)} \mathcal{R}_\theta\|_{\text{MSE}} + \xi_3 \|\delta_{\kappa(i)} \mathcal{T}_\theta\|_{\text{MSE}}, \tag{10}$$

where $\kappa$ is the map introduced in Eq. (6). By using the priority scoring in Eq. (10), the buffer can sample various experiences that are useful to MaCN for both long-term and short-term viewpoints. The detailed descriptions of our framework MaPER are provided in Algorithm 1.

Finally, we remark that the computational overhead by MaPER is negligible, compared to the vanilla PER. Suppose that a critic network has a linear layer with $N$ hidden units as the final layer. Then one can compute the difference in the number of parameters between the MaCN and the original critic network: $N(1 + |S|)$, where $S$ is a space of (embedded) states in critic networks. Notice that the number of additional parameters depends only on the final hidden units in the critic network. Therefore, computing costs for the additional parameters is minor compared to computing costs for both the total parameters in all networks and prioritized experience sampling.

## 3 EXPERIMENT

In this section, we conduct experiments to answer the following questions:

- Can the proposed method enhance the performances of diverse off-policy RL algorithms having $Q$-networks in various environments?
- What attributes to the success of our method are the most?

To answer them, we first describe our experimental setup and show the main results against relevant baselines to show that our method is generally applicable to RL algorithms including critic networks and experience replays, and largely improves their performance. Then we perform an ablation study of our method to analyze the most crucial components of it. In all experiments, the solid lines and shaded regions in all experimental results represent the mean and standard deviations across five runs with random seeds.

### 3.1 EXPERIMENTAL SETUP

**RL algorithms.** We validate the effectiveness of MaPER with the following algorithms: Soft Actor-Critic (SAC) (Haarnoja et al., 2018a), Twin Delayed Deep Deterministic policy gradient (TD3), Rainbow (Hessel et al., 2018), and Model-based Policy Optimization (MBPO) (Janner et al., 2019). Here, we apply data-efficient Rainbow (van Hasselt et al., 2019) since its sample efficiency is dramatically higher than the original. When applying our method to given algorithms, we do not alter the original hyper-parameters to show that our method effectively improves the base algorithm's performance without hyperparameter tuning.

**Environments.** For continuous control tasks, we consider not only MuJoCo environments (Todorov et al., 2012), which have been frequently used to validate Many RL algorithms, but also PyBullet Gymperium[1], which are free implementations of the original MuJoCo environments. Other free environments in the OpenAI (Brockman et al., 2016) are also considered. For discrete control tasks, we validate our method on Atari games. Finally, we consider sparse reward variants of Pendulum-v0 and HumanoidPyBulletEnv to show that our method is also effective on them. We provide the details of the environments we used for the experiments in the **supplementary material**.

**Sampling methods.** We also compare the performance of our method against the following experience replays to the original algorithms. The first is Experience Replay with Uniform Sampling at Random (RANDOM). The second is the vanilla PER (Schaul et al., 2016), which is rule-based prioritized sampling of the transitions based on TD-errors. The third is experience replay with Likelihood-free Importance Weights (LfIW) (Sinha et al., 2020), a learning-based prioritizing method, which predicts the importance of each experience.

**Reward shaping methods.** We also consider reward shaping methods to show that MaPER can seamlessly be combined with them. Since there are various proposed reward shaping methods (Pathak et al., 2017; Stadie et al., 2015; Devlin & Kudenko, 2012; Zou et al., 2019; Hu et al., 2020; Wiewiora et al., 2003; Ng et al., 1999), we combine MaPER to some of them. The first is Curiosity-driven Exploration (CE) (Pathak et al., 2017). This method uses additional intrinsic rewards to promote exploration: $r_t^i = \frac{\eta}{2}\|\widehat{\phi}(s_{t+1}) - \phi(s_{t+1})\|$, where $\phi(s_t)$ is a feature vector and $\widehat{\phi}(s_t)$ is a predicted feature vector for $s_t$, respectively. The second is Incentivizing Exploration (IE) (Stadie et al., 2015). This method uses a modified reward $r_t + \mathcal{V}(s_t, a_t)$ with $\mathcal{V}(s_t, a_t) = \frac{e_t}{CT \max_{t' \leq t} e_{t'}}$, where $C$ is a decay rate constant, and $e_t = \|\zeta(s_{t+1}) - \mathcal{M}(\zeta(s_t), a_t)\|_{\text{MSE}}$. Here, $\mathcal{M}$ is a learnable network by $e_t$ for predicting embedding $\zeta(s_t)$ of $s_t$. In a similar manner to the original results, we implement additional networks for reward shaping independently of the other networks.

The **supplementary material** provides more detailed descriptions of our experimental settings, including the details about the hyper-parameters and implementations.

### 3.2 MAIN EXPERIMENTAL RESULTS

**MfRL**. Figure 2 shows the learning curves of SAC and TD3 on three PyBullet environments and BipdalWalkerHardcore-v3. Our MaPER significantly and consistently outperforms baselines in

---

[1]https://github.com/benelot/pybullet-gym

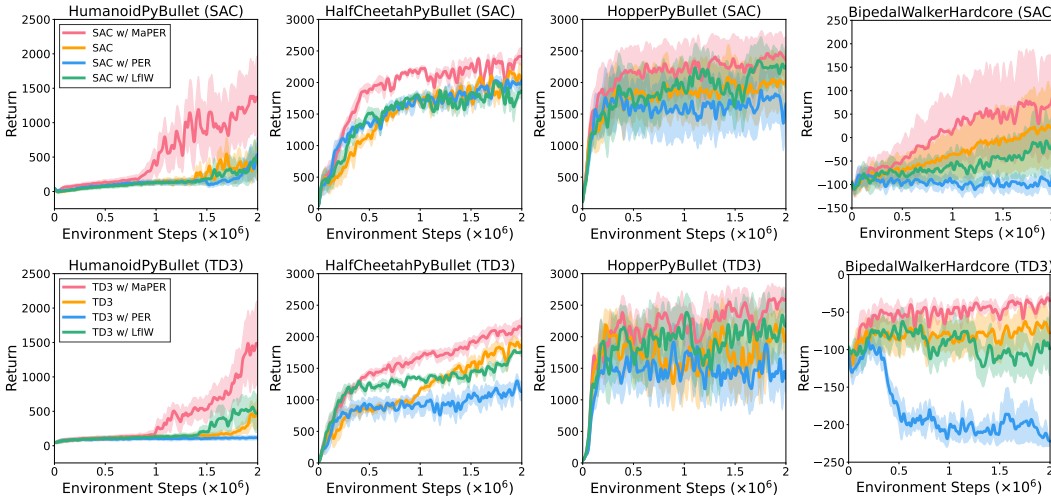

Figure 2: Learning curves of the baseline off-policy model-free algorithms with various sampling methods. Here, the algorithms with our method outperform the original ones. The solid lines and shaded regions represent the mean and standard deviations by ten evaluation across ten runs with random seeds.

Table 1: Average of cumulative rewards under Rainbow on Atari Environments after 2M training steps by 10 evaluations across five runs with random seeds. One can observe that Rainbow with MaPER outperforms Rainbow without MaPER on most tasks. Here, we used the same hyperparameters of Rainbow as (van Hasselt et al., 2019; Lee et al., 2020a) except for the total timesteps (0.1M → 2M) to observe the much longer time behavior (see Figure C.1 in the supplementary material for learning curves).

| Game | Return w/ MaPER | Return w/o MaPER |
|---|---|---|
| Alien | **1635.6** (±390.0) | 1351.4 (±296.3) |
| Amidar | **482.6** (±103.4) | 427.5 (±163.6) |
| Assault | **866.4** (±144.7) | 508.1 (±60.1) |
| Asterix | **1281.6** (±215.1) | 1122.6 (±202.6) |
| BankHeist | **1035.0** (±134.3) | 542.6 (±330.2) |
| Boxing | **2.2** (±2.5) | 0.9 (±4.6) |
| Breakout | **7.1** (±2.3) | 6.5 (±2.3) |
| ChopperCommand | **1565.8** (±985.9) | 1358.3 (±525.9) |
| DemonAttack | **1870.6** (±954.0) | 1281.3 (±658.1) |
| Freeway | **32.9** (±0.6) | 32.4 (±0.3) |
| Frostbite | **4712.0** (±712.9) | 3147.4 (±686.3) |
| Gopher | **714.8** (±30.0) | 578.6 (±192.9) |
| Hero | **14954.4** (±2328.5) | 14088.6 (± 3495.3) |
| Jamesbond | **831.2** (±39.8) | 515.3 (± 120.3) |
| Kangaroo | **8385.9** (±2149.3) | 6120.2 (± 2944.9) |
| Krull | **4846.8** (±769.1) | 3981.1 (± 557.8) |
| KungFuMaster | **11483.6** (±1804.1) | 10808.2 (± 1668.1) |
| MsPacman | **1823.3** (±268.5) | 1791.9 (± 445.8) |
| Pong | 19.8 (±1.2) | 19.8 (± 2.3) |
| PrivateEye | 98.3 (±0.0) | **120.0** (± 40.0) |
| Qbert | **9364.3** (±1974.4) | 8845.9 (± 2826.9) |
| RoadRunner | **17911.1** (±7441.5) | 15088.6 (± 2853.6) |
| Seaquest | **742.8** (± 126.0) | 630.4 (± 81.6) |
| UpNDown | **8465.6** (± 6867.0) | 7383.5 (± 2320.9) |

all tested cases. Next, since we focus on improving $Q$-learning, we validate our method with data-efficient Rainbow (van Hasselt et al., 2019), which is one of the state-of-the-art $Q$-learning frameworks. Table 1 shows the performance ratio of the Rainbow with MaPER to the original Rainbow. One can observe that Rainbow with MaPER achieves overwhelmingly higher performance over the base Rainbow on most Atari games.

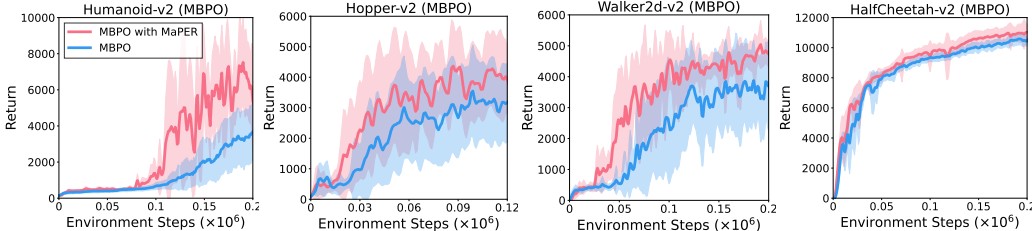

Figure 3: The average cumulative rewards obtained on continuous control tasks using MBPO and MBPO + MaPER. Our method significantly improves the performance of MBPO. The solid lines and shaded regions denote the mean and standard deviations by one evaluation across five runs with random seeds.

**MbRL**. Figure 3 shows the learning curves of the MBPO with and without MaPER on the MuJoCo environments. We observe that MBPO with MaPER consistently outperforms the vanilla MBPO. In particular, similarly to MfRL on HumanoidPyBullet (Figure 2), MaPER obtains the largest performance gain on Humanoid-v2, which has the highest-dimensional state and action spaces among tasks in MuJoCo environments, which may require larger number of samples for accurate $Q$-value estimation. These impressive results with MBPO further show MaPER's versatility and effectiveness.

**Sparse reward**. We experiment MaPER on the sparse reward tasks by combining it with existing reward shaping methods (see Table 2), to show its effectiveness on them. Although MaPER does not use specifically designed rewards to solve for sparse reward tasks, it obtains reasonable performance gain over the original SAC, due to stimulated exploration while MaCN's additional learning to predict the environment behavior with MaPER. We further observe that combining MaPER and reward shaping methods results in considerable performance improvements over the reward shaping alone.

We provide additional experimental results in the **supplementary material.**

Table 2: Average of cumulative rewards under SAC on sparse reward environments by 10 evaluations after 2M training steps across five runs with random seeds. Combining MaPER (our method) and reward shaping method results in the best performances.

| Method | Sparse-Pendulum | Sparse-HumanoidPyBullet |
|---|---|---|
| Original | 00.00 ($\pm$00.00) | 00.00 ($\pm$00.00) |
| IE | 32.89 ($\pm$40.27) | 00.08 ($\pm$00.08) |
| CE | 49.19 ($\pm$40.17) | 00.14 ($\pm$00.06) |
| MaPER (our method) | 16.30 ($\pm$32.61) | 00.17 ($\pm$00.13) |
| IE+MaPER | **82.69** ($\pm$01.35) | 00.53 ($\pm$00.54) |
| CE+MaPER | 81.12 ($\pm$02.39) | **00.83** ($\pm$00.45) |

### 3.3 ABLATION STUDY

We analyze what components in our method are crucial to its improvement of RL's performances.

**Effectiveness of MaPER.** Figure 4.(a) compares the learning curves of SAC with MaCN, but different sampling methods to verify the effectiveness of MaPER. There is a considerably large gap between MaPER and MaCN alone. Moreover, one can observe that MaCN with LfIW and PER are inferior to RANDOM. The main reason is that these two sampling methods focus on sampling transitions that are beneficial in updating the $Q$-network only (i.e., TD-errors and importance weights for the $Q$-loss), but estimation of the $Q$-value is only a single component of MaCN $\mathcal{C}_\theta$. So, it is suboptimal in improving $\mathcal{C}_\theta$. Yet, MaPER seeks to improve the estimation of the reward, state, and $Q$-values, so that it samples diverse and effective experiences. As a result, it outperforms other sampling methods under SAC with MaCN.

**Effectiveness of learning the environment behavior.** Next, we analyze the effectiveness of the environment estimation in MaPER. To verify it, we examine the performance of MaPER without the reward or transition estimators. Figure 4.(b) shows the learning curves of each case. It shows that learning both of the environment estimators are useful to improve the performance.

**Effectiveness of parameter sharing.** To show that computing priority scores by additionally considering model-errors is effective in itself, we considered networks separate from the original

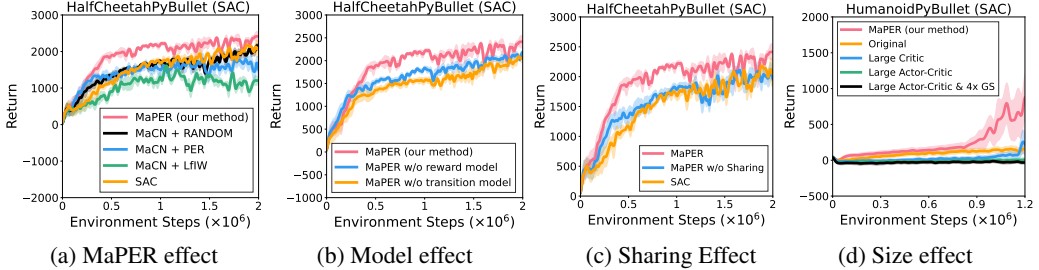

Figure 4: Ablation study. (a): Different prioritizing methods under MaCN. We observe that MaPER outperforms other prioritizing methods under MaCN. (b): Different estimators in MaPER. One can observe that estimating both reward and transition is the most effective. (c): Learning curves of MaPER with and without sharing networks. Here, networks, which predict rewards and next states, used the same number of hidden units with the same architecture as the original critic networks under SAC. (d): Network size effect under SAC. Large networks have hidden units (2500, 2500), respectively. Simply increasing the number of parameters and gradient steps (GS) leads to marginal or worse performance. The solid lines and shaded regions represent the mean and standard deviations by ten evaluation across ten runs with random seeds.

critic networks under SAC in Figure 4.(c), i.e., networks without parameter sharing. The separated networks have the same number of hidden units with the same architecture as the original critic networks under SAC. One can observe that the final performance of MaPER without parameter sharing is on a par with MaPER with parameter sharing, i.e., with MaCN.

**Effectiveness of increased parameters.** One may suspect that if MaPER achieves improved performance due to the increased size of the $Q$-network, with additional estimators. To show that this is not the case, we use the original SAC with its hidden layers dramatically increased, from $(400, 300)$ to $(2500, 2500)$ for both the $Q$-network and the policy network. Figure 4.(d) shows the effect of changes by learning curves. We observe that increasing the hidden layer size alone does not improve the performance and may even lead to the performance degeneration. Further, one can observe that increasing gradient steps cannot change the tendency. Therefore, the improvements of our method is not simply coming from the increased size of hidden layers.

## 4 RELATED WORK

**Model-based RL.** Although our approach is fundamentally different from Model-based RL (MbRL) approaches which generate virtual experiences to train agents by planning, we briefly introduce some of them because we want to verify our method in MbRL. Thus far, various MbRL methods (Kurutach et al., 2018; Luo et al., 2018; Clavera et al., 2018; Janner et al., 2019; Rajeswaran et al., 2020; Clavera et al., 2020; Silver et al., 2017; Schrittwieser et al., 2020; Hafner et al., 2020a; Kaiser et al., 2019) have been proposed, but the common strategy across them is to first learn the environment models and use them to generate fictitious experiences for learning a policy. Due to its ability to generate transitions, MbRL's sample efficiency is remarkable on certain tasks although much larger computing costs are generally needed.

**Off-policy model-free RL.** Among $Q$-learning algorithms, Rainbow (Hessel et al., 2018), which combines various techniques for deep $Q$-learning (Mnih et al., 2015), is one of the state-of-the-art methods. Specially, data-efficient Rainbow (van Hasselt et al., 2019) outperforms the original one and other existing methods. In the case of the policy-based methods, Twin Delayed Deep Deterministic policy gradient (TD3) (Fujimoto et al., 2018), Soft Actor-Critic (SAC) (Haarnoja et al., 2018a) and their variants (Wang & Ross, 2019; Yang et al., 2021) are the state-of-the art methods that are frequently used as baselines. Here, TD3 employs double $Q$-networks, target policy smoothing, and different frequencies to update a policy and $Q$-networks. SAC also adopts double $Q$-learning and utilizes the entropy measure of an agent policy to the reward to encourage the exploration of the agent. They are also utilized as a component of MbRL (Janner et al., 2019).

**Experience replay.** Prioritized Experience Replay (PER) and its variants are one of the most frequently used methods to sample relatively important transitions from the replay buffer for $Q$-learning (Hessel et al., 2018; Schaul et al., 2016; Wang & Ross, 2019; Brittain et al., 2019). For instance, Based on PER, Brittain et al. (2019) proposed a method to increases the priorities of previous

transitions resulting in the important transitions. The effect of increasing capacity and downweighting old transition in the buffer has been also studied (Fedus et al., 2020). Wang et al. (2015); Zhang & Sutton (2015) also stuided the importance of utilizing the newest experiences for RL's learning. Of course, instead of TD-error, a different metric can be also used to PER, e.g., the expected return (Isele & Cosgun, 2018; Jaderberg et al., 2016) which is also obtainable from critic networks like TD-error. Apart from PER and its variants, Novati & Koumoutsakos (2019) proposed a replay method in which policy updates are penalized if samples are far-policy. Sun et al. (2020) proposed attentive experience replay, which compute similarity between on-policy and past experiences to sample most similar experiences. By the way, some results (Andrychowicz et al., 2017; Fang et al., 2019b;a; Liu et al., 2019a) designed experience replay methods which relabel experiences to stimulate exploration for sparse reward environments. Experience replay is also important to initialize states for planning. Pan et al. (2019; 2020) applied hill climbing to generate initial states to generate virtual experiences (the search control) on the value function estimate. Recently, learning-based experience replay methods, which utilizes neural networks to generate priority scores, have been proposed (Zha et al., 2019; Sinha et al., 2020). Specifically Zha et al. (2019) designed a network, whose input is the concatenation of reward, timestep, and TD-error, which computes the Bernoulli distribution's probability on each experience to select suitable experiences in DDPG (Lillicrap et al., 2016) algorithm on continuous action tasks. Sinha et al. (2020) proposed a learnable likelihood-free density ratio estimator between on-policy and off-policy experiences, whose input is a tuple of action and state, to compute the importance weights. Finally, Oh et al. (2021) designed a neural sampler, whose input consists of $Q$-value, TD-error, transition's element, and timestep, which computes the relative importance between experiences to sample diverse and useful experiences. However, to the best of our knowledge, all learnable features are only obtained from critic networks even in learning-based methods.

**Shared weights.** Shared weights are frequently used in multi-agent RL frameworks (Rashid et al., 2018; Papoudakis et al., 2019), mostly for learning the policy networks. Further, sharing weights between the actor and critic networks is a standard approach (Mnih et al., 2016; Silver et al., 2017; Schulman et al., 2017), since it allows to learn representations that can contribute to both of them. To resolve practical issues with shared weights (e.g., balancing objectives), some methods propose to decouple the actor and the critic networks (Raileanu & Fergus, 2021; Cobbe et al., 2020). Moreover, shared weights between model and $Q$-value estimators have been used to enhance representation learning for pixel-image observations (Lin et al., 2019; Agarwal et al., 2021; Stooke et al., 2021; Lee et al., 2020b; Schrittwieser et al., 2020; Hessel et al., 2021; Hafner et al., 2020b; Srinivas et al., 2020; Guo et al., 2020; Lee et al., 2019) or reward for stimulating exploration (Pathak et al., 2017; Shelhamer et al., 2016). However, to the best of our knowledge, none of the existing works attempted to share weights across the critics and environment models to enhance experience replay.

## 5  DISCUSSION AND CONCLUSION

We propose Model-augmented Prioritized Experience Replay (MaPER) which computes priority scores of past experiences based on the model estimation errors, as well as the TD-errors. To implement this, we employ a Model-augmented Critic Network (MaCN) that estimates not only the $Q$-value but also the environment behavior via weight sharing, where environment estimations are useful to predict $Q$-values. By providing experiences with high model errors at the early stage of training, MaPER brings a curriculum learning effect for predicting $Q$-values well at later stages.

The advantages of our proposed method are as follows. First, MaPER dramatically increases sample efficiency of state-of-the-art algorithms: SAC, TD3, Rainbow, and MBPO since it largely alleviates the underestimation or overestimation of the value with the conventional $Q$-learning. Second, it is simple to implement and is applicable to any model-free and -based algorithms that utilize $Q$-networks. Third, the computational cost of our algorithms is almost the same as the original algorithms. Finally, it is effective in various environments, including ones with the sparse rewards.

Despite the advantages above, unfortunately, there is no theoretical analysis of under what conditions and domains our method is effective. As a result, in certain complicated environments with high-dimensional state and action spaces for which the model training is extremely difficult, our method may achieve lower performance than that of the model-free baselines. In such environments, MaPER may require an excessive amount of interactions with the environments to train the model estimators. However, in such extreme cases, all-model based methods will be similarly ineffective.

## ETHICS STATEMENT

Agents in reinforcement learning have potential threats of damaging human property and life. Indeed, RL agents trained under a wrongly designed reward may choose actions that sacrifice humans to increase their returns. Besides, even if agents select suitable actions in a simulation, it is not easy to guarantee they will behave well in the real world. In the future, RL agents may also cause many workers to lose their jobs due to the overwhelming performance of RL agents in various areas. Accordingly, we claim that RL should be developed with consideration of how to tackle the issues. For instance, various methods for safety (Tessler et al., 2019; Pinto et al., 2017; Chow et al., 2018; Wachi & Sui, 2020; Alshiekh et al., 2018; Cheng et al., 2019) should be proposed in parallel of methods for improving the sample efficiency of RL.

## REPRODUCIBILITY STATEMENT

We describe the implementation details for experiments in Appendix B. We also provide our code.

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

# Supplementary Material:

## Model-augmented Prioritized Experience Replay

### ORGANIZATION

This supplementary material provides descriptions of the materials that the main paper does not cover and additional experimental results to help readers understand our method. The document is organized as follows.

**Appendix A** - We describe environments, which belong to MuJoCo, PyBullet, OpenAI Gym, and Atari environments, considered in both the main paper and the supplementary material.

**Appendix B** - We provide the parameters for model-free RL, model-based RL, and reward shaping experiments conducted in this paper.

**Appendix C** - We provide additional experimental results about Atari and MuJoCo environments and ablation study to validate MaPER (our method) in this appendix.

## A    ENVIRONMENT DESCRIPTION

### A.1    MUJOCO ENVIRONMENTS

Multi-Joint dynamics with Contact (MuJoCo)[2] (Todorov et al., 2012) is a physics engine for robot simulations supported by OpenAI Gym[3]. The MuJoCo environments are currently some of the most widely used toolkits for developing and comparing RL algorithms. In MuJoCo environments, Reinforcement Learning (RL) agents should learn a policy to control the joints (action) for achieving a goal, i.e., cumulative rewards. The observation of each environment includes information about the angular velocity and position of the robot's joints. This paper has considered the following MuJoCo environments.

**Humanoid-v2** is an environment where RL agents control a three-dimensional bipedal robot on the ground. Agents should learn how to walk forward without falling over.

**HalfCheetah-v2** is an environment where RL agents control a two-dimensional cheetah robot for learning sprint.

**Hopper-v2** is an environment where RL agents control a one-legged robot. The robot receives a high return if it hops forward as soon as possible without falling over.

**Walker2d-v2** is an environment where RL agents control a robot with two-dimensional bipedal legs to walk. Learning to quick walking without failure guarantees a high return.

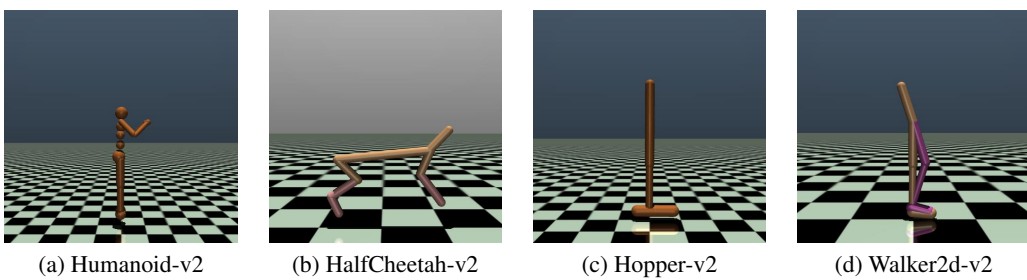

| (a) Humanoid-v2 | (b) HalfCheetah-v2 | (c) Hopper-v2 | (d) Walker2d-v2 |

Figure A.1: MuJoCo environments.

---

[2]http://www.mujoco.org/
[3]https://gym.openai.com/

## A.2 PYBULLET ENVIRONMENTS

We also use PyBullet environments[4], which are the open-source implementations of the MuJoCo environments, for our experiments for anyone's reproducibility. Besides, compared to MuJoCo environments, these environments are more difficult due to the more realistic setting, e.g., adding energy cost.

**HumanoidPyBullet(Env-v0)** is an environment in which RL agents control a three-dimensional bipedal robot to walk quickly without falling over.

**HalfCheetahPyBullet(Env-v0)** is an environment in which RL agents control a two-dimensional cheetah robot for learning sprint.

**HopperPyBullet(Env-v0)** is an environment in which RL agents control a two-dimensional one-legged robot hop-forward quickly without falling over.

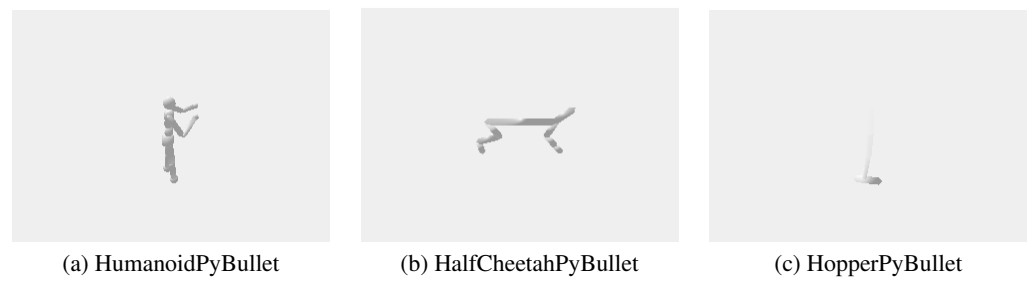

(a) HumanoidPyBullet     (b) HalfCheetahPyBullet     (c) HopperPyBullet

Figure A.2: PyBullet environments.

## A.3 OTHER ENVIRONMENTS

**BipedalWalkerHardcore(-v3)** is a variant of BipedalWalker-v3 in OpenAI Gym. The robot's objective is to move forward as far as possible while escaping many obstacles which do not appear in the original.

**Sparse-Pendulum** is an environment which objective is to balance a rod in the upright position as long as possible. It is a variant of Pendulum-v0 that OpenAI Gym supports. To make it sparser, we impose the following condition: The pendulum begins to receive +1 reward if maintaining the rod in the upright position more than 100 steps continuously.

**Sparse-HumanoidPyBullet** is a variant of the HumanoidPyBullet(Env-v0) environment in which the robot receives +1 at every 200 timesteps if the robot has not fallen over. The other things are entirely equivalent to the original.

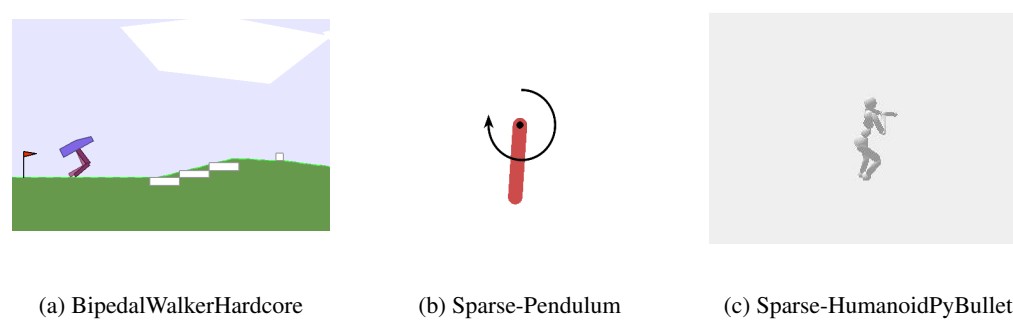

(a) BipedalWalkerHardcore     (b) Sparse-Pendulum     (c) Sparse-HumanoidPyBullet

Figure A.3: Other Gym environments.

---

[4]https://github.com/benelot/pybullet-gym/

Table A.1: Dimensions of observation and action spaces for continuous control environments.

| Environment | Observation space | Action space | Horizon |
|---|---|---|---|
| Humanoid(-v2) | $\mathbb{R}^{376}$ | $[-1,1]^{17}$ | 1000 |
| Hopper(-v0) | $\mathbb{R}^{17}$ | $[-1,1]^{6}$ | 1000 |
| Walker2d(-v2) | $\mathbb{R}^{17}$ | $[-1,1]^{6}$ | 1000 |
| HalfCheetah(-v2) | $\mathbb{R}^{17}$ | $[-1,1]^{6}$ | 1000 |
| HumanoidPybullet(Env-v0) | $\mathbb{R}^{44}$ | $[-1,1]^{17}$ | 1000 |
| HalfCheetahPybullet(Env-v0) | $\mathbb{R}^{26}$ | $[-1,1]^{6}$ | 1000 |
| HopperPybullet(Env-v0) | $\mathbb{R}^{15}$ | $[-1,1]^{3}$ | 1000 |
| BipedalWalkerHardcore(-v3) | $\mathbb{R}^{24}$ | $[-1,1]^{4}$ | 2000 |
| Sparse-Pendulum | $\mathbb{R}^{3}$ | $[-1,1]^{1}$ | 200 |
| Sparse-HumanoidPyBullet | $\mathbb{R}^{44}$ | $[-1,1]^{17}$ | 1000 |

Table A.1 shows the observation and action spaces and the maximum environment steps for each episode (horizon) in MuJoCo, Pybullet, and other OpenAI Gym environments in this paper. Here, $\mathbb{R}$ and $[-1,1]$ denote sets of real numbers and those between 0 and 1, respectively.

## A.4 DISCRETE CONTROL ENVIRONMENTS

We now describe the Atari environments that we consider in the main article and supplementary material. The objective of RL agents here is to learn a policy of discrete actions (buttons) by observing the screen (RGB) to get high cumulative rewards defined for each game.

**Alien** is an environment where a player should destroy all alien eggs in the RGB screen while escaping aliens.

**Amidar** is an environment similar to MsPacman. In this environment, agents control a monkey in a fixed rectilinear lattice to eat pellets as much as possible while escaping chasing masters.

**Assault** is an environment where a player controls a spaceship, whose objective is to eliminate the enemies.

**Asterix** is an environment where a player controls a tornado. Its objective is to eat hamburgers on the screen with avoiding dynamites.

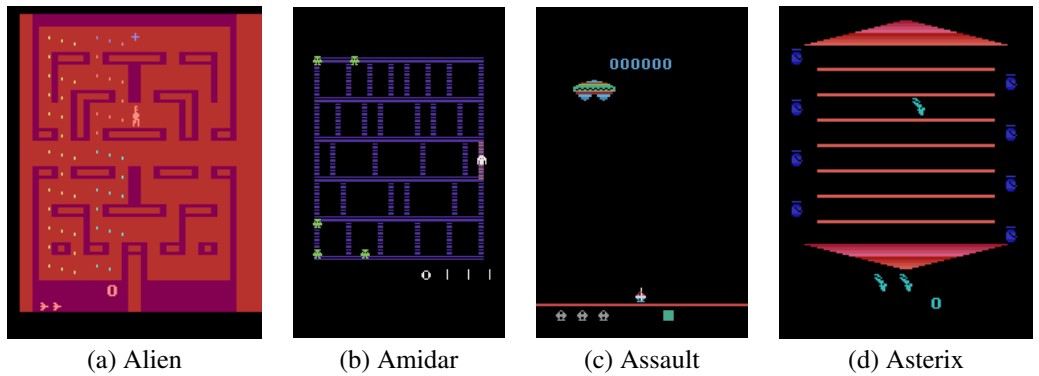

|             |             |             |             |
|:-----------:|:-----------:|:-----------:|:-----------:|
| (a) Alien   | (b) Amidar  | (c) Assault | (d) Asterix |

Figure A.4: Atari Games: Alien, Amidar, Assault, and Asterix.

**BankHeist** is an environment where a player controls a robber whose objective is to rob banks as many as possible while avoiding the police in maze-like cities.

**Boxing** is an environment about the sport of boxing. There are two boxers with a top-down view, and RL agents should control one of them. They get one point if their punches hit from a long distance and two points if their punches hit from a close range. A match is over after two minutes, or 100 punch hits to the opponent.

**Breakout** is an environment where a player should destroy all bricks by controlling a paddle that makes a ball rebound.

**ChopperCommand** is an environment in which a player operates a helicopter in a desert. The helicopter should destroy all enemy aircraft and helicopters while protecting a convoy of trucks.

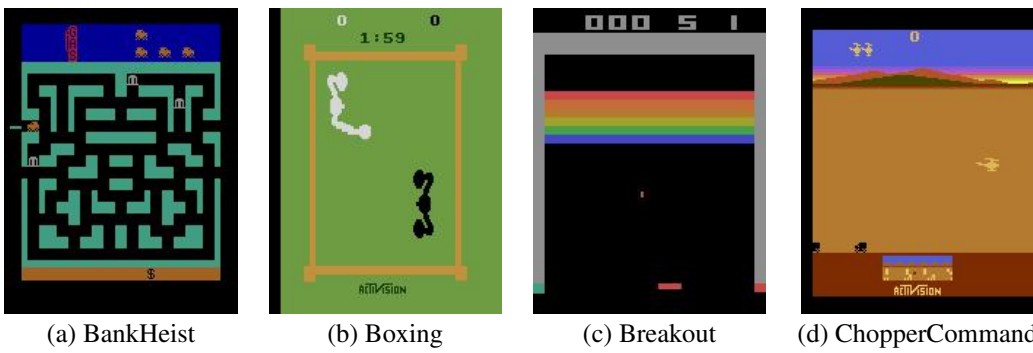

|               |            |              |                    |
|:-------------:|:----------:|:------------:|:------------------:|
| (a) BankHeist | (b) Boxing | (c) Breakout | (d) ChopperCommand |

Figure A.5: Atari Games: BankHeist, Boxing, Breakout, and ChopperCommand.

**DemonAttack** is an environment in which a player controls a guardian who should kill demons that attacks from above.

**Freeway** is an environment where a player controls chickens to run across a ten-lane highway with traffic. They are only allowed to move up or down. The objective is to get across as possible as they can until two minutes.

**Frostbite** is an environment where a player controls a man who should collect ice blocks to make his igloo, whose objective is to collect 15 ice blocks while avoiding some opponents, e.g., crabs and birds.

**Gopher** is an environment where a player controls a farmer who should protect three carrots from a gopher.

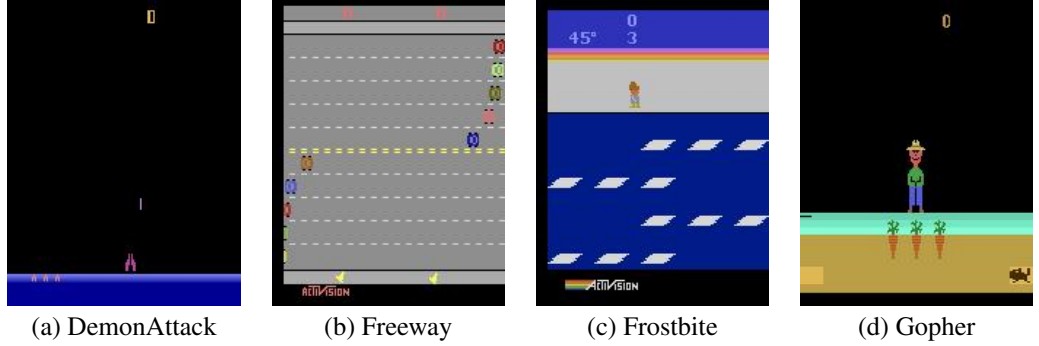

(a) DemonAttack  (b) Freeway  (c) Frostbite  (d) Gopher

Figure A.6: Atari Games: DemonAttack, Freeway, Frostbite, and Gopher.

**Hero** is an environment in which a player should control a man who has a backpack-mounted helicopter unit for rescuing a miner trapped at the bottom.

**Jamesbond** is an environment where a player controls a vehicle whose objective is to move forward while avoiding and attacking enemies.

**Kangaroo** is an environment where a player controls a mother kangaroo, whose objective is to rescue her son while climbing.

**Krull** is an environment that controls a player should complete stages, which are central parts of the film which name is the same.

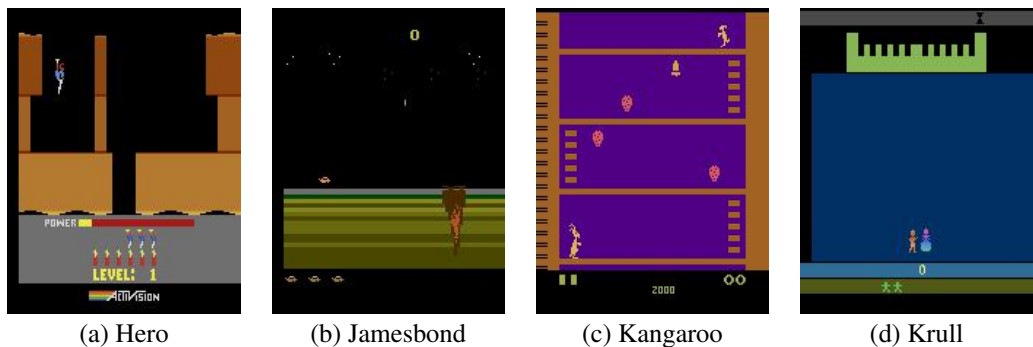

(a) Hero  (b) Jamesbond  (c) Kangaroo  (d) Krull

Figure A.7: Atari Games: Hero, Jamesbond, Kangaroo, and Krull.

**KungFuMaster** is an environment where a player should control a fighter to save his girlfriend. He can use two types of attacks (punch and kick) and move/crunch/jump actions.

**MsPacman** is an environment where a player controls a Pacman in given mazes for eating pellets as much as possible while avoiding chasing masters. The Pacman loses one life if it contacts with monsters.

**Pong** is an environment about table tennis. RL agents control an in-game paddle to hit a ball back and forth. The objective is to gain 11 points before the opponent. The agents earn each point when the opponent fails to return the ball.

**PrivateEye** is an environment where a player controls a private eye. It requires action, adventure, and memorization. For solving five cases, the private eye should find and return items to suitable places.

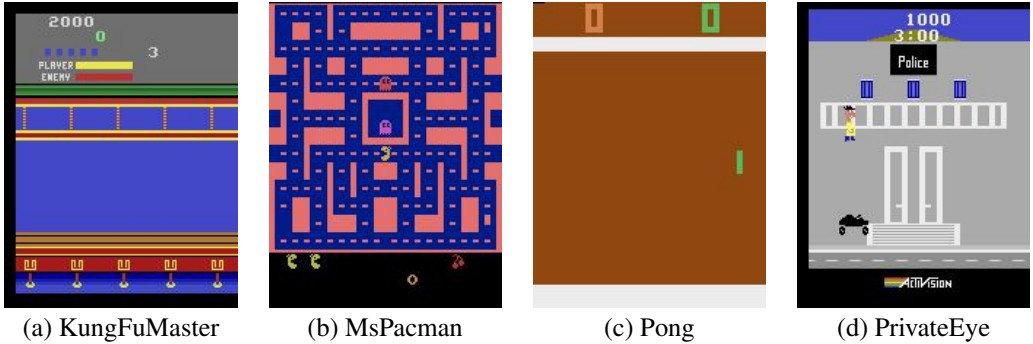

(a) KungFuMaster    (b) MsPacman    (c) Pong    (d) PrivateEye

Figure A.8: Atari Games: KungFuMaster, MsPacman, Pong, and PrivateEye.

**Qbert** is an environment where a player controls a character under a pyramid made of 28 cubes. The character should change the color of all cubes while avoiding obstacles and enemies.

**RoadRunner** is an environment where a player controls a roadrunner (chaparral bird). The roadrunner runs to the left on the road. RL agents should pick up bird seeds while avoiding chasing coyotes and obstacles such as cars.

**Seaquest** is an environment where a player controls a submarine, whose objective is to rescue divers while attacking enemies with missiles.

**UpNDown** is an environment where a player should control a purple dune buggy for collecting flags while avoiding other cars.

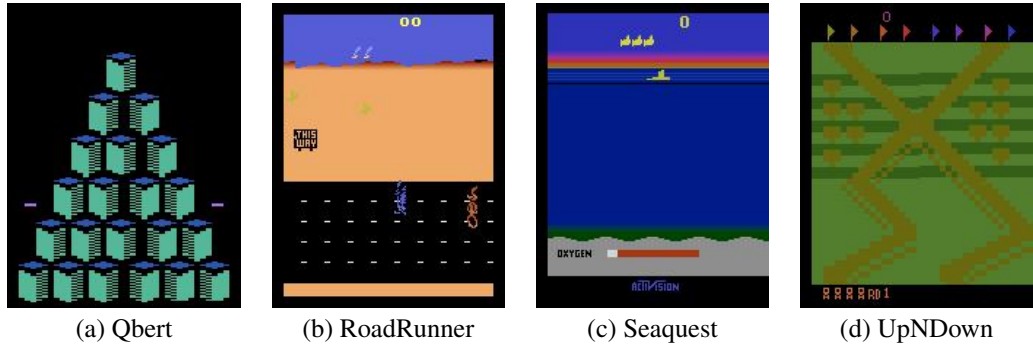

(a) Qbert    (b) RoadRunner    (c) Seaquest    (d) UpNDown

Figure A.9: Atari Games: Jamesbond, Kangaroo, Krull, and Seaquest.

# B TRAINING DETAILS

Table B.1: Parameters for Model-free Reinforcement Learning experiments.

| Parameter | Value |
|---|---|
| **Shared** | |
| Batch size | 256 (SAC), 100 (TD3), 32 (Rainbow) |
| Buffer size | $10^6$ |
| Target smoothing coefficient ($\tau$) for soft update | 0.02 |
| Initial PER exponents $(\alpha, \beta)^5$ | (0.7, 0.4) (SAC/TD3), (0.5, 0.4) (Rainbow) |
| Discount factor for the agent reward ($\gamma$) | 0.98 (SAC/TD3), 0.99 (Rainbow) |
| Number of initial random actions | $10,000$ (SAC/TD3) $5,000$ (Rainbow) |
| Optimizer | Adam (Kingma & Ba, 2014) |
| Nonlinearity | ReLU |
| Replay period | 64 (SAC/TD3), 1 (Rainbow) |
| Gradient step | 64 (SAC/TD3), 1 (Rainbow) |
| **MaCN** | |
| Temperature for dynamic coefficients (Liu et al., 2019b) | 50 |
| **Likelihood-free Importance Weights** | |
| Temperature for weights | 5 |
| Hidden units of networks | 256, 256 |
| Fast replay buffer size | $10^4$ |
| **TD3** | |
| Hidden units of MLP networks | 400, 300 |
| Learning rate | $10^{-3}$ |
| Policy update frequency | 2 |
| Gaussian action and target noises | 0.1, 0.2 |
| Target noise clip | 0.5 |
| Target network update | Soft update with interval 1 |
| **SAC** | |
| Hidden units of MLP networks | 400, 300 |
| Learning rate | $7.3 \times 10^{-4}$ |
| Target entropy | $-\dim A$ ($A$ is action space) |
| Soft update with interval | 1 |
| **Rainbow** | |
| Observation down-sampling for Atari RGB | $84 \times 84$ with grey-scaling |
| CNN channels for Atari environments | 32, 64 |
| CNN filter size for Atari environments | $5 \times 5, 5 \times 5$ |
| CNN stride for Atari environments | 5, 5 |
| Action repetitions and Frame stack | 4 |
| Reward clipping | True ($[-1, 1]$) |
| Terminal on loss of life | True |
| Max frames per episode | $1.08 \times 10^5$ |
| Target network update | Hard update (every 2,000 updates) |
| Support of $Q$-distribution | 51 |
| $\epsilon$ for Adam optimizer | $1.5 \times 10^4$ |
| Learning Rate | $10^{-4}$ |
| Max gradient norm | 10 |
| Noisy nets parameter | 0.1 |
| Multi-step return length | 20 |
| $Q$-network's hidden units per layer | 256 |

Table B.1 includes the parameters that are used for the experiments under Model-free Reinforcement Learning (MfRL) in this paper. Since there are few benchmark results of PyBullet environments compared to MuJoCo environments, we followed parameters in (Raffin & Stulp, 2020) for SAC and

---

[5] $\beta$ increases to 1.0 by the rule $\beta = 0.4\eta + 1.0(1 - \eta)$, where $\eta$ = the current step/the maximum steps.

Table B.2: Parameters for MBPO (Janner et al., 2019) experiments.

| Parameter | Value |
|---|---|
| Epochs | 400 (HalfCheetah), 300 (Walker2d, Humanoid), or 125 (Hopper) |
| Environments steps per epoch | 1,000 |
| Model rollouts per environment step | 400 |
| Ensemble size | 7 |
| Hidden units of MLP networks | 4 hidden layers of size 400 (Humanoid), or 4 hidden layers of size 200 (otherwise) |
| Policy updates per environment steps | 40 (HalfCheetah) or 20 (otherwise) |
| Model horizon | $1 \rightarrow 15$ over epochs $20 \rightarrow 100$ (Hopper), $1 \rightarrow 25$ over epochs $20 \rightarrow 300$ (Humanoid), or 1 (otherwise) |
| Initial PER exponents $(\alpha, \beta)$ | $(1.0, 0.4)$ (Humanoid), $(0.5, 1.0)$ (otherwise) |
| Temperature for dynamic coefficients | 50 |

TD3. In the case of MBPO, to use MaPER in MBPO, we modified the critic networks in SAC, which is the core of MBPO, to MaCN for adding the model-learning components. Notice that MBPO utilizes two buffers to collect true and virtual experiences, respectively. Then MaPER is used to compute both true and virtual experiences' priority scores, which are updated whenever updating SAC networks. In the case Rainbow, we followed parameters used in (van Hasselt et al., 2019; Lee et al., 2020a), but we increased the total environment steps to observe tendencies over longer time steps. Notice that MaCN only requires the temperature value of dynamic coefficients (Liu et al., 2019b) for multi-task learning with Eq. /(5). We implemented MaPER in Rainbow such that that MaCN's transition estimator learns to predict deep representations from the CNN layers for the current and next RGB arrays as input and output, instead of taking the RGB array as input and output. Moreover, since Rainbow (van Hasselt et al., 2019) uses the multi-step return with length 20 to compute the TD-error instead of the reward, we also use it instead of the estimated reward. Table B.2 shows parameters for applying MBPO (Janner et al., 2019), which is one of the state-of-the-art model-based RLs. We basically employed parameters introduced in (Janner et al., 2019) for MBPO experiments. Here, only PER's exponents $(\alpha, \beta)$ and temperature for dynamic coefficients are additionally considered. Finally, Table B.3 provides parameters for the reward shaping. Here, $|\mathcal{O}|$ represents the dimension of observations. **We provide a source code for the main experiments.**

Table B.3: Parameters for the reward shaping (Pathak et al., 2017; Stadie et al., 2015) experiments.

| Parameter | Value |
|---|---|
| Curiosity-driven Exploration | |
| Architecture of feature network $\phi$ for observation $\mathcal{O}$ | MLP with 2 hidden layers: $(\lceil|\mathcal{O}|/2\rceil, \lceil|\mathcal{O}|/4\rceil)$ |
| Inverse model architecture | MLP with 2 hidden layers: $(256, 256)$ |
| Importance weight of the inverse model loss against the forward model loss | 0.2 |
| Importance weight of the policy gradient loss signal against the learning the intrinsic reward | 0.1 |
| Incentivizing Exploration | |
| Architecture of encoder $\sigma$ | MLP with 2 hidden layers: $(\lceil|\mathcal{O}|/2\rceil, \lceil|\mathcal{O}|/4\rceil)$ |
| Architecture of predictive Model $\mathcal{M}$ | MLP with 2 hidden layers: $(\lceil|\mathcal{O}|/2\rceil, \lceil|\mathcal{O}|/2\rceil)$ |
| Decay constant $C$ | 1.0 |
| The current environment step | $T$ |

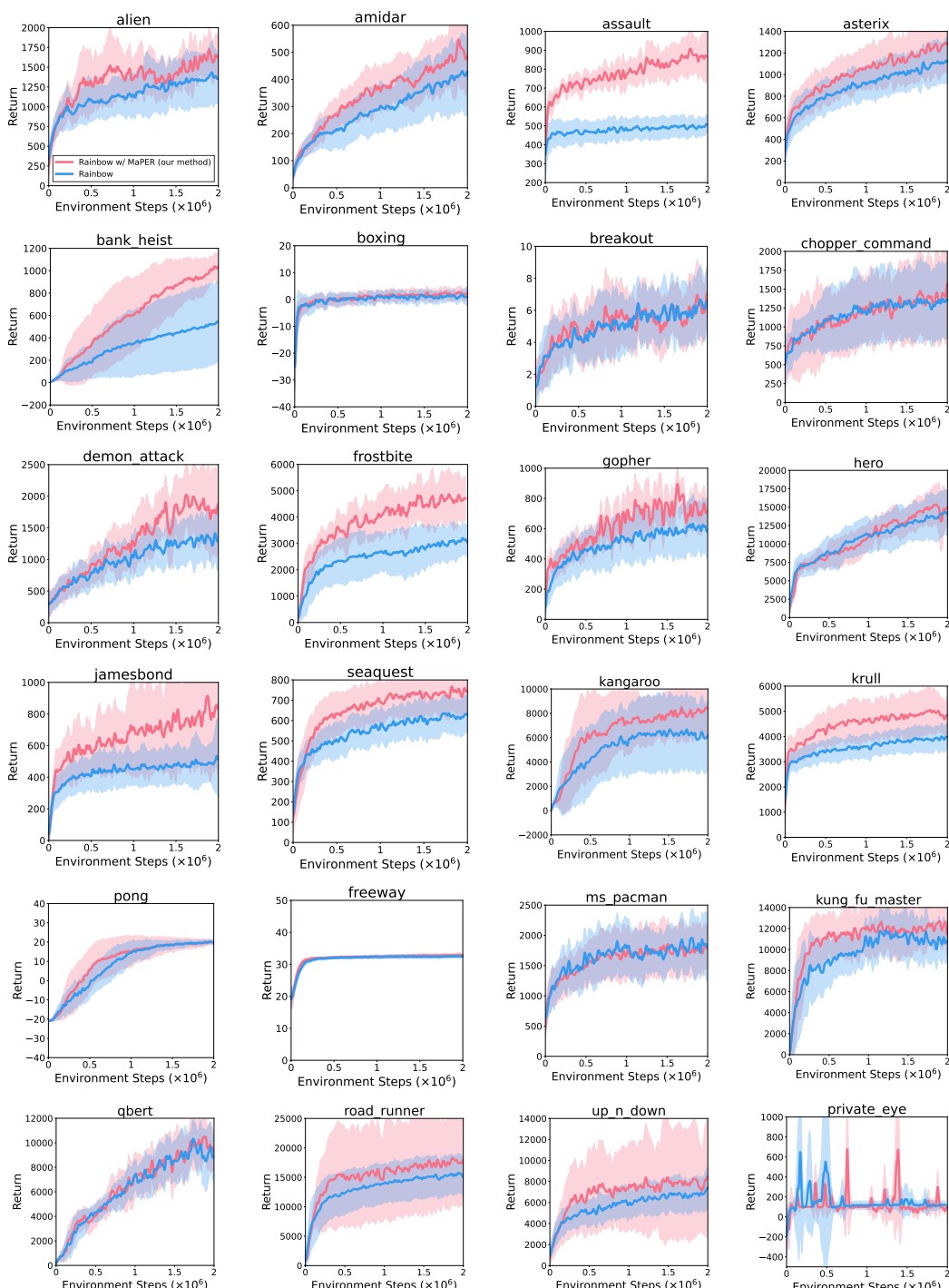

Figure C.1: Learning curves of Rainbow on Atari games. The solid line and shaded regions represent the mean and standard deviation by 10 evaluations, respectively, across five runs with random seeds.

## C   ADDITIONAL EXPERIMENTAL RESULTS

**Learning curves on Atari games.** Figure C.1 shows Rainbow's learning curves of the average of performances for 2M environment steps with and without MaPER on Atari games. Rainbow with MaPER consistently outperforms the original Rainbow in most experimental results.

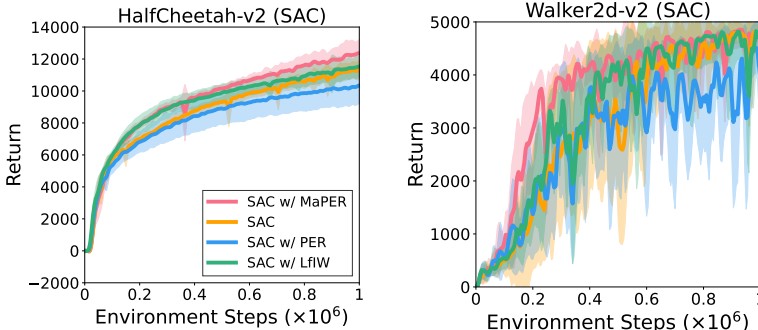

Figure C.2: Learning curves of SAC on MuJoCo environments. The solid line and shaded regions represent the mean and standard deviation by one evaluation, respectively, across five runs with random seeds.

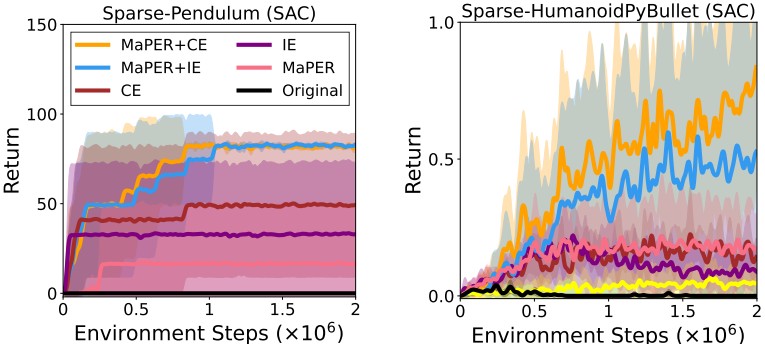

Figure C.3: Learning curves of SAC on sparse reward environments. The solid line and shaded regions represent the mean and standard deviation by one evaluation, respectively, across five runs with random seeds.

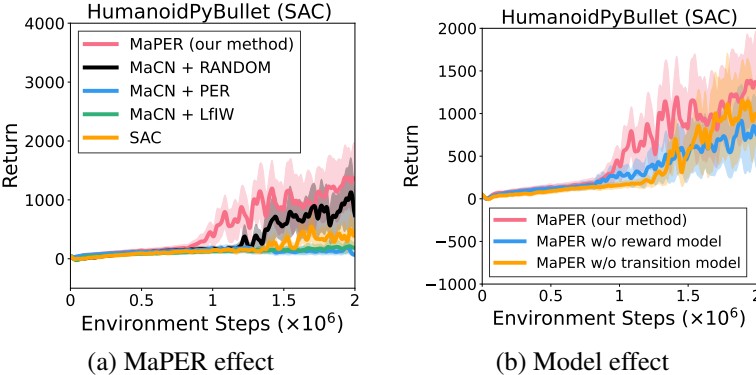

(a) MaPER effect          (b) Model effect

Figure C.4: Ablation Study. (a): Different prioritizing methods with MaCN. We observe that MaPER outperforms other prioritizing methods under MaCN. (b): Different estimators in MaPER. One can observe that estimating both reward and transition is the most effective. The solid line and shaded regions represent the mean and standard deviation by ten evaluation, respectively, across ten runs with random seeds.

Table C.1: Average of cumulative rewards under SAC on sparse reward by 10 evaluations after 2M training steps across five runs with random seeds. The values in parentheses denote standard deviations. MaPER + Real Reward means that the TD-error equation (see Eq. (4)) uses the real rewards instead of model rewards.

| Sparse-HumanoidPyBullet (SAC) | | | |
|---|---|---|---|
| Steps | MaPER | MaPER + Real Reward | Original |
| 1.0M | 00.17 ($\pm$00.18) | 0.03 ($\pm$00.05) | 00.00 ($\pm$00.00) |
| 1.5M | 00.20 ($\pm$00.20) | 0.03 ($\pm$00.06) | 00.00 ($\pm$00.00) |
| 2.0M | 00.17 ($\pm$00.13) | 0.04 ($\pm$00.06) | 00.00 ($\pm$00.00) |

The learning curves of SAC on MuJoCo environments are shown in Figure C.2. We observe that SAC with MaPER outperforms SAC with other sampling methods. Figure C.3 shows the learning curves of SAC (Haarnoja et al., 2018a) on the sparse reward environments in the main article (see Table B.3), which reveals that the original algorithm could not learn a policy for returns on these environments. We observe that combining MaPER with reward shaping methods guarantees the best results compared to other methods, including reward shaping methods solely.

Under SAC, we computed the training wall clock times between MaPER and the vanilla PER in Table C.2. One can observe that the increased wall clock-times are marginal as we discussed in the final paragraph in Section 2.

Table C.2: The training wall-clock time between MaPER and PER under SAC. We computed the elapsed time for train networks from sampling across five runs with random seeds.

| Task | MaPER (hours) | PER (hours) | Ratio (MaPER/PER) |
|---|---|---|---|
| HalfCheetahPyBullet | 5.89 | 5.50 | 1.07 |
| HopperPyBullet | 5.95 | 5.84 | 1.02 |
| HumanoidPyBullet | 6.34 | 5.93 | 1.07 |

Figure C.4 shows the effectiveness of both MaPER and learning the environment behavior, on HumanoidPyBullet. Comparing Figure 4 which considers HalfCheetahPyBullet, one can observe the same tendency, i.e., MaPER is necessary to improve the performance. Table C.1 shows the performance of SAC with MaPER on Sparse-HumanoidPyBullet (SAC) at time steps 1.0M, 1.5, and 2.0M, respectively. The table shows that using model rewards to compute TD-errors in MACN stimulates agents' exploration.

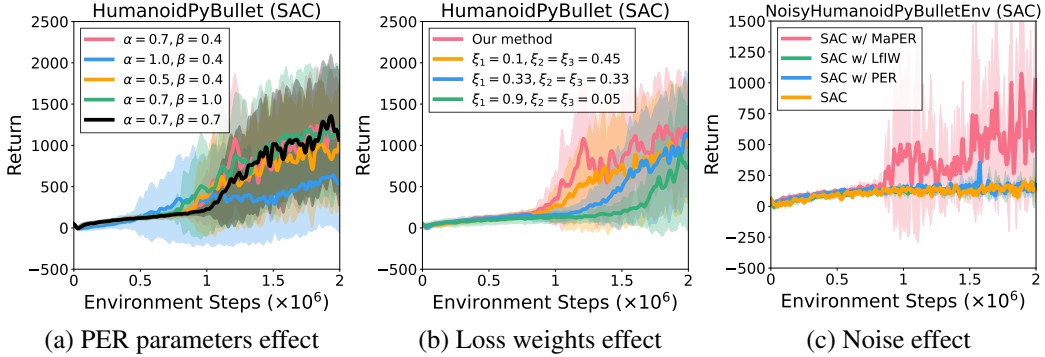

(a) PER parameters effect     (b) Loss weights effect     (c) Noise effect

Figure C.5: Ablation Study. (a): Different hyper-parameters $\alpha$ and $\beta$ of PER. (b): Comparison between dynamic weights (our method) and fixed weights of the loss (5) in MaPER under SAC. Some hyper-parameters degrade the performance of MaPER. (c): Learning curves on HumanoidPyBullet with Gaussian noise $\mathcal{N}(0, 0.01)$ to rewards and next states. MaPER considerably outperforms other methods. The solid line and shaded regions represent the mean and standard deviation by one evaluation, respectively, across five runs with random seeds.

One can observe the sensitivity of MaPER with respect to PER's parameters $(\alpha, \beta)$ and coefficients in the loss (5) in Figure C.5.(a)-(b). In most cases, final results are similar, but some hyper-parameters degrade the performance of MaPER. To confirm the robustness of our method on an noise environment, we considered HumanoidPyBullet with Gaussian noise $\mathcal{N}(0, 0.01)$ to rewards and next states in Figure C.5.(c). MaPER considerably outperforms other methods since MaPER provides samples for improving the representation of inputs of critic networks.

Transitions of Sparse-Pendulum while training under SAC are displayed in Figure C.6. In both cases (MaPER and PER), the agents did not receive any reward. A position of the pendulum, which is trained by samples based only on TD-errors, is in downward. However, the pendulum trained by MaPER explores various states to receive a reward. Finally, Table C.3 shows holistic performances for different domains, which are derived in our paper. We normalized each method's performance by that of MaPER for each domain. We believe that the considerably large gap between PER and MaPER in continuous control tasks under MfRL results from the actor-critic architecture (SAC and TD3). Compared to $Q$-learning, the advantage of the actor-critic architecture is to derive faster convergence (Konda & Tsitsiklis, 2000). Conversely speaking, if the performance of critic networks is consistently poor, then the policy networks can quickly converge to a poorly behaving policy so that learning

will be stuck. We think that as the prediction of $Q$-values with MaPER quickly converges to the true returns (see Figure 1.(c) in the manuscript), the policy networks can be well-trained. Therefore, the large gap shows that how to improve the critic networks is crucial to not only $Q$-learning but also modern algorithms having the actor-critic architecture.

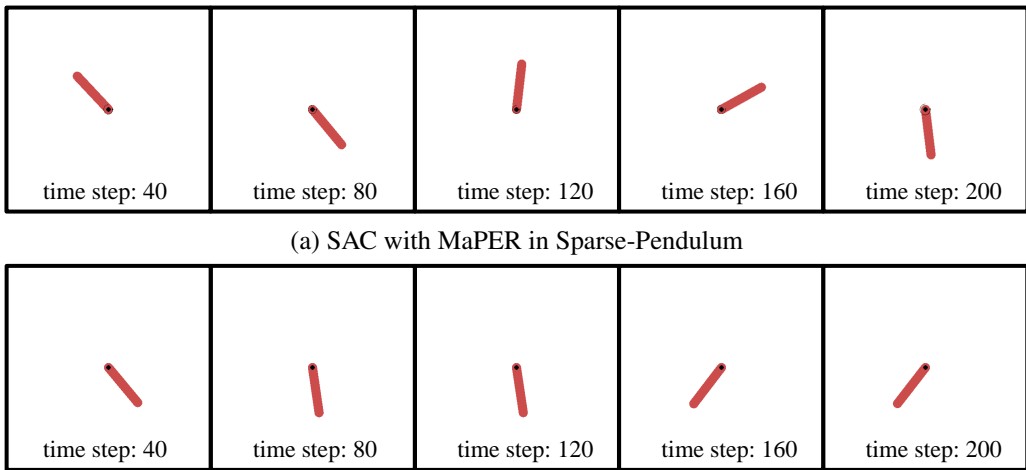

(a) SAC with MaPER in Sparse-Pendulum

(b) SAC with PER in Sparse-Pendulum

Figure C.6: Transitions on Sparse-Pendulum while training under SAC with MaPER and PER, respectively, from 50,000 environment steps. Although both MaPER and PER did not find any reward, SAC with MaPER have tried various states of transitions, i.e., the rod in downward, upright, and horizontal positions.

Table C.3: Holistic performances for each domain. Here, in each domain, we normalized each method's performance based on MaPER's. One can observe that MaPER considerably outperforms other methods.

| Domain | MaPER | RANDOM | PER | LfIW |
|---|---|---|---|---|
| MfRL (continuous control tasks) | **1.00** | 0.68 | 0.27 | 0.68 |
| MfRL (discrete control tasks) | **1.00** | N/A | 0.75 | N/A |
| MbRL (continuous control tasks) | **1.00** | 0.80 | N/A | N/A |

