# OpenReview forum: "Model-augmented Prioritized Experience Replay"
_ICLR.cc/2022/Conference — ICLR 2022 Poster_

### Official Review · Reviewer_K56x · 2021-10-29

**Correctness:** 3
**Technical Novelty And Significance:** 2
**Empirical Novelty And Significance:** 2
**Recommendation:** 6
**Confidence:** 5

**Main Review:**

*Main Review

Pros:
1. The experiment section is very complete.
There’s a model-free part that considers some of representative algorithms such as SAC, TD3,
and there's a model-based part that compares against MBPO.
And experiments are implemented on both the continuous robotics tasks and atari games.

2. The details of the paper are adequate.
The main paper and the appendix section provides a lot of the details of the algorithm,
making it quite reproducible.

3. Sufficient related work.

4. Compared to the baselines, the authors show that the proposed algorithm has better performance.

5. The code is released.

Cons:
1. The algorithm is relatively simple.
And there’s no theoretical evidence that supports the proposed algorithms,
and the inspiration from this paper is not clear.
Adding the dynamics terms in the priority seems pretty random.

2. The sensitivity.
It is not clear how sensitive the parameters in the algorithm are.
For example $\xi_1$, $\xi_2$, $\xi_2$ in equation (5) are three variables i think that could be quite unstable and sensitive. It would be interesting to see if the algorithm only works on 1 set of hyper-parameters, or it is in general a very robust algorithm



**Summary Of The Paper:**


*Summary Of The Paper
Model-augmented Prioritized Experience Replay
In this paper, the authors consider using the dynamics prediction error in the priority calculation.
The priority is then used to decide the probability for each sample during training.
The method is applicable to both model-based and model-free setting


**Summary Of The Review:**


*Summary Of The Review

I am quite worried about the fact that the algorithm is looking quite simple.
But the results are looking good.

---

> ### Author Response · Authors · 2021-11-18
> **Response to Reviwer K56x**
>
> We deeply appreciate your efforts and insightful comments to improve the manuscript. We respond to each of your comments one-by-one in what follows. In the revised manuscript, we have marked the revisions with “green.”
>
> *******
>
> **[Q1]** The algorithm is relatively simple. And there’s no theoretical evidence that supports the proposed algorithms, and the inspiration from this paper is not clear. Adding the dynamics terms in the priority seems pretty random.
>
> **[A1]** We believe that the simplicity of our algorithm is rather its strength. Moreover, our inspiration originates from the definition of Q-value, which is the expectation of returns under a given policy (see Eq. (3.13) in [1]), so adding the dynamic terms in the priority makes sense as follows. If we completely understand the distribution of the rewards and transitions on a tuple of action and state, then Q-values are theoretically computable. In other words, we thought learning to predict  environment models is meaningful to improve the representation for predicting Q-values.  Besides, the prediction of the reward and next state can be confirmed after the one-step unlike Q-values, which can be confirmed after the multi-steps. Accordingly, the prediction of environment models is relatively easier than that of Q-values. To conclude, we expected the curriculum learning effect by providing samples which are additionally useful for learning models, to predict $Q$-values. It is why we designed MaPER.
>
>
> [1] Sutton, Richard S., and Andrew G. Barto. Reinforcement learning: An introduction. MIT press, 2018.
>
> ******************
>
> **[Q2]** It would be interesting to see if the algorithm only works on 1 set of hyper-parameters, or it is in general a very robust algorithm
>
> **[A2]** Following your suggestion, we have attached Figure C.5.(a)-(b), which show the sensitivity analysis with respect to PER parameters and coefficients of the multi-objective loss, respectively, in the supplementary material. In the figures, one can observe MaPER is not too sensitive with the hyper-parameters.

---

### Official Review · Reviewer_rZAM · 2021-10-31

**Correctness:** 4
**Technical Novelty And Significance:** 3
**Empirical Novelty And Significance:** 3
**Recommendation:** 8
**Confidence:** 4

**Main Review:**

Strengths
=========
+ The paper is well-written overall and is easy to digest.
+ Through and strong experimental results
+ Simple technique that is generalizable

Room for Improvement
=========
- Dive deep into the Private Eye domain. Why didn't the proposed approach work? I love to see that inside the paper.
- While the additional complexity of the algorithm was discussed (N(1+|S)), it would be really great if they have compared the performance of their algorithm against the others, where the x-axis is wall close-time.
- Provide a more holistic number across all domains for better sending the message out. For example, normalizing the return between 0-1 and then provide the average across all domains.

Details
=========
- Perhaps remove the acronyms in the abstract.
- Add line number to your Algorithm.
- I did not follow the priority set on line 5 of Algorithm. Can you elaborate ?
- Figure 2, Table 2, Figure 4 captions: "across five runs with randoms" => "across five runs with random seeds".
- Please avoid reintroducing the acronyms. Example MaPER in sec 3.1
- "Since LfIW was only validated it on" => remove it
- Page 8: "Besides, the performance between" => "Besides, the performance difference between"
- What was unique about private eye? Any insights there?


**Summary Of The Paper:**

Authors introduced Model-augmented Prioritized Experience Replay (MaPER) as a variation to the canonical prioritized sweeping. The main difference is that they extended the critic network on its last layer to also predict the reward and transition model. Then they changed the prioritization of sample for replay to also include the error in the reward and transition model, in addition to the TD-error. Experimental results in various MuJoC domains showed the performance boost introduced by MaPER on top of existing state-of-the-art techniques. Furthermore Authors carried an ablation study, investigation the impact of 1) incorporating the MaPER while all methods used the same network 2) incorporating various error values for prioritization, and 3) increasing the network size.

**Summary Of The Review:**

I liked this paper. The proposed approach is simple and generic (more sparse reward domains should be investigated for their final claim to be substantiated) yet practical. The experimental results are comprehensive and I think it would be a great value to RL practitioners.

---

> ### Author Response · Authors · 2021-11-18
> **Response to Reviwer rZAM**
>
> We deeply appreciate your efforts and insightful comments to improve the manuscript. We respond to each of your comments one-by-one in what follows. In the revised manuscript, we have marked the revisions with “green.”
>
> *******
>
> **[Q1]** I did not follow the priority set on line 5 of Algorithm. Can you elaborate?
>
> **[A1]** Line 5 describes inserting a transition with initializing its priority score by the maximum priority score in the buffer (as like the vanilla PER).
>
> *******
>
> **[Q2]** What was unique about the private eye?
>
> **[A2]** We believe that there is nothing special in the private eye. In this task, both MaPER and PER perform very poorly (see their learning curves in Figure C.1 in the supplementary material) due to the sparsity of reward. In Table 1, it seems that MaPER is worse than PER in the private eye, but their scores are too low and comparisons are less meaningful.
>
> *******
>
> **[Q3]** Compare the performance of wall-clock times.
>
> **[A3]** Following your suggestion, we provide the table below showing the training wall-clock time between MaPER and PER under SAC. As expected, the difference in training wall-clock times is marginal. We have added the table in the revised manuscript (see Table 3 in Section 3.3).
>
> \begin{array}{cccc}
> \hline
> \text{Task} & \text{MaPER (hours)}       & \text{PER (hours)} & \text{Ratio (MaPER/PER)}\newline
> \hline
> \text{HalfCheetahPyBullet}  & 5.89 & 5.50 & 1.07  \newline
> \text{HopperPyBullet} & 5.95 & 5.84 & 1.02         \newline
> \text{HumanoidPyBullet} & 6.34 & 5.93 & 1.07
> \newline
> \hline
> \end{array}
> *******
>
> **[Q4]** Provide a more holistic number across all domains for better sending the message out.
>
> **[A4]** Following your suggestion, we provide the table below showing holistic performances for different domains: 1) MfRL on continuous tasks, 2) MfRL on discrete  control tasks and 3) MbRL on continuous control tasks. We normalized each method’s performance by that of MaPER for each domain. We believe that the considerably large gap between PER and MaPER in continuous control tasks under MfRL results from the actor-critic architecture (SAC and TD3). Compared to $Q$-learning, the advantage of the actor-critic architecture is to derive faster convergence [1]. Conversely speaking, if the performance of critic networks is consistently poor, then the policy networks can quickly converge to a poorly behaving policy so that learning will be stuck. We think that as the prediction of $Q$-values with MaPER quickly converges to the true returns (see Figure 1.(c) in the manuscript), the policy networks can be well-trained. Therefore, the large gap shows that how to improve the critic networks is crucial to not only $Q$-learning but also modern algorithms having the actor-critic architecture.
>
> \begin{array}{c|cccc}
> \hline
> \text{Domain} & \text{MaPER} & \text{RANDOM}   & \text{PER}    & \text{LfIW}     \newline \hline
> \text{MfRL (continuous control tasks)}    & \textbf{1.00}    & 0.68 & 0.27 & 0.68 \newline
> \text{MfRL (discrete control tasks)}     &  \textbf{1.00}   & \text{N/A}      & 0.75   & \text{N/A}      \newline
> \text{MbRL (continuous control tasks)}   &  \textbf{1.00}     & 0.80     & \text{N/A}    & \text{N/A}      \newline \hline
> \end{array}
>
> [1] Konda, Vijay R., and John N. Tsitsiklis. "Actor-critic algorithms." Advances in neural information processing systems. 2000.
> *******
>
> **[Q5]** Minor comments:
> * Perhaps remove the acronyms in the abstract.
> * Add line number to your Algorithm.
> * Figure 2, Table 2, Figure 4 captions: "across five runs with randoms" $\rightarrow$ "across five runs with random seeds".
> * Please avoid reintroducing the acronyms. Example MaPER in sec 3.1
> * "Since LfIW was only validated it on" $\rightarrow$ remove it
> * Page 8: "Besides, the performance between" $\rightarrow$ "Besides, the performance difference between"
>
> **[A5]** We appreciate the reviewer's careful reading. We have reflected the above in the revised manuscript.

---

### Official Review · Reviewer_noEm · 2021-11-02

**Correctness:** 2
**Technical Novelty And Significance:** 2
**Empirical Novelty And Significance:** 3
**Recommendation:** 5
**Confidence:** 3

**Main Review:**

Strength:

1. The paper focuses on an important topic: what should be the sampling distribution for ER? And the proposed method seems new.

2. The presentation is reasonably clear.

3. The empirical results show that the proposed method is promising.

Weaknesses:

1. Although the proposed method is supported by some intuitions, it does not have a solid motivation.

In page 1, it is said “learning to predict … is difficult, so sampling based on TD-error is far from optimal.” What do you mean by “short term”? And why sampling from TD error is far from optimal? Any reference?

The effectiveness of PER is questionable …. The authors never introduce what these limitations of PER are. I think the limitations of PER should be reviewed in more detail in this work and should be explained how the proposed method can address these limitations.

An additional note, reducing TD errors faster does not tell us anything about the sample efficiency of learning performance. I personally have some experience with it. Reducing TD errors fast can still have a bad policy. For example, when your buffer only covers a small subset of the state space.  Hence, I do not understand the effect of showing figure 1(b).

According to your intuition, during early learning, samples with large model errors would get sampled more frequently. The authors should provide some insight (either empirically or theoretically) to explain why such sampling distribution is preferred. I am doubting it has some exploration effect. There is also work saying that TD error magnitude can be used for encouraging exploration (Smart Exploration in Reinforcement Learning using Absolute Temporal Difference Errors by Clement et al), so it seems it is not a problem to use TD errors during the early learning stage. You might compare the sample space coverages during the early learning stage between your algorithm and regular PER.

The intuition is not persuasive. During early learning, how do you know the model error plays a dominant role? It seems domain-dependent. And, it is also doubtful whether a representation can describe the dynamics imply it can predict Q values. They can be quite different. In some environments, learning a model can be much more challenging (e.g., high dimensional), it is possible that the model error term always dominates the priority.

2.  The contribution is incremental. Basically, it adds a model error term to the TD error.

3. What if the transition probability p(s’|s,a) is not Gaussian, how do you measure the error?

The three losses can have very different numerical scales, how to handle this?

4. The experiments are not sufficient.

a. What is the parameter sweep for each algorithm? PER is sensitive to the alpha, beta parameter in (8), (9)

b. Do you ever try to use a separate NN for model learning (but still use the same priority as you defined)? This can help identify if the auxiliary tasks are useful or not.

c. It would be interesting to see how the method works when increasing uncertainty in 1) reward function 2) transition function. This can help identify if the benefit comes from uncertainty measurement when calculating the model errors.

Missing citations:
1. The credit of ER should go to Self-Improving Reactive Agents Based On
Reinforcement Learning, Planning and Teaching by Lin, 1992.

2. Page 2, “we remark … ” There are indeed works improving RL’s experience replay by employing model learning. Please see the SGLD-based sampling in MBRL (e.g., Frequency-based search control by Pan et al.) I believe there is a category of work along this direction.

**Summary Of The Paper:**

The paper proposes a new method for prioritizing experiences used in the prioritized experience replay (PER) method. The proposed approach is simple: the critic network also learns the reward function and transition function. The two errors (absolute value) are added to the absolute TD error to calculate the priorities in the PER method. The authors provide some intuitions to their method. Experiments on both Mujoco and Atari environments are presented to show the method’s effectiveness.

**Summary Of The Review:**

Please see main review.

---

> ### Author Response · Authors · 2021-11-18
> **Response to Reviwer noEm (2/2)**
>
> **[Q7]** The case of Non-Gaussian transition probability:
>
> **[A7]** We appreciate the reviewer's useful comment. In the case of different distributions from the Gaussian, one can utilize the inductive bias about suitable loss equations for given distributions. For instance, in the Bernoulli distribution case, we can utilize the binary cross entropy loss to measure the error of model prediction components in MaCN.
>
> ************
>
> **[Q8]** MaPER’s sensitivity with respect to $\alpha$ and $\beta$
>
> **[A8]** To analyze the sensitivity as you pointed out, we have added Figure C.5.(a) which shows how our method works for different PER parameters $\alpha$ and $\beta$. One can observe that MaPER is not too sensitive with the hyper-parameters, but some parameters degrade the performance of MaPER; we suspect that the poor performance when $\alpha=1.0$ results from the too high bias of sampling.
>
> **[Q9]** Do you ever try to use a separate NN for model learning?
>
> **[A9]** Yes, but using MaCN is more effective; Figure 4.(c), which we have added in the revised manuscript,shows learning curves with and without parameter sharing. From the figure, one can conclude that calculating priority scores additionally with model-errors is quite meaningful to improve the performance because auxiliary learnable features can be used to sample various meaningful experiences for training compared to the single feature (TD-error).
>
> **[Q10]** Performance in noisy environment
>
> **[A10]** Following your suggestion, we have conducted experiments to verify the effectiveness of MaPER in a noisy environment, where the Gaussian noise $\mathcal{N}(0, 0.01)$ are added to rewards and next states from HumanoidPyBulletEnv. We have added Figure C.5.(c) in the supplementary material. From the figure, one can observe that other methods except MaPER could not learn to receive high cumulative rewards. We believe that the improved representation of a tuple of action and state leads to MaPER’s outperforming performance.
>
> *******
>
> **[Q11]** How to handle Different numerical scales of multi-objective losses.
>
> **[A11]** To handle it, we considered **the dynamic weight average** [1], which is one of the methods to handle multi-objective learning, as we mentioned in Eq. (5) in the manuscript. This method observes previous multi-objective losses to determine a relative weight for each term. If some of the multi-objective losses are saturated, then their weights will be relatively small.
>
> [1] Liu, Shikun, Edward Johns, and Andrew J. Davison. "End-to-end multi-task learning with attention." Proceedings of the IEEE/CVF Conference on Computer Vision and Pattern Recognition. 2019.
>
> ************
>
> **[Q12]** Missing citations: 1) The credit of ER; 2) SGLD-based sampling in MBRL vs MaPER
>
> **[A12]** We deeply appreciate the reviewer for introducing meaningful references, which we missed. We added the first reference [1], which is credited for experience replay, on the first paragraph of Section 1: Introduction in the manuscript. The methods [2, 3] based on the SGLD-based sampling in model-based RL are also interesting, but they are about how to initialize states based on **value networks** for generating virtual experiences. These methods apply the hill climbing approach which utilizes the gradient ascent based on some metrics related to value networks (please see 7.(a)-7.(b) in [3]). From those states chosen by the hill climbing approach, **the model components are only used to generate virtual experiences.** Besides, as 7.(a)-7.(b) in [3], the requirement for computing the second derivative of the metrics may be impractical for high-dimensional states unlike our method. Nevertheless, contributions of those methods are meaningful for experience replay, so we added [6, 7] in Section 4: Related Work.
> References:
>
> [1] Lin, Long-Ji. "Self-improving reactive agents based on reinforcement learning, planning and teaching." Machine learning 8.3-4 (1992): 293-321.
>
> [2] Pan, Yangchen, et al. "Hill Climbing on Value Estimates for Search-control in Dyna."
>
> [3] Pan, Yangchen, Jincheng Mei, and Amir-massoud Farahmand. "Frequency-based Search-control in Dyna." arXiv preprint arXiv:2002.05822 (2020).

---

> ### Author Response · Authors · 2021-11-18
> **Response to Reviwer noEm (1/2)**
>
> We deeply appreciate your efforts and insightful comments to improve the manuscript. We respond to each of your comments one-by-one in what follows. In the revised manuscript, we have marked the revisions with “green.”
>
> *************
>
> **[Q1]** What do you mean by “short term” in Section 1: Introduction?
>
> **[A1]** By “short term'', we intended to mention that learning to predict $Q$-values is generally **difficult with a small number of steps in a given environment**. To make it more clear, we revised the 3rd paragraph of Section 1 as follows: "However, measuring $Q$-values requires to predict the expectation of returns, which can be obtained after taking multiple steps; thus, learning to predict $Q$-values generally requires a large number of interactions with an environment".
>
> *************
>
> **[Q2]** Why sampling from TD error is far from optimal? Any reference?
>
> **[A2]** It is known that PER (sampling from TD error) is often ineffective even compared to the uniform random sampling in some tasks, e.g., please see task 'defender' of Figure 6 in [1], Figure 2-3 in [2], and Figure 2 in [3] (see Section 1’s third paragraph in our manuscript). In particular, [2] even reported that experiences with low TD-errors could be helpful to train agents in some tasks. To resolve PER's poor performance issue, we employed the model components additionally for experience replay. As a result, we showed that our method is effective in both continuous and discrete control tasks under various RL algorithms.
>
> [1] Hessel, Matteo, et al. "Rainbow: Combining improvements in deep reinforcement learning." Thirty-second AAAI conference on artificial intelligence. 2018.
>
> [2] Zha, Daochen, et al. "Experience Replay Optimization."
>
> [3] Sinha, Samarth, et al. "Experience replay with likelihood-free importance weights." arXiv preprint arXiv:2006.13169 (2020).
>
> *******
>
> **[Q3]** Reducing TD errors faster does not tell us anything about the sample efficiency of learning performance.
>
> **[A3]**
>
> - We agree that reducing TD-errors does not explain everything in RL. However, we believe that it is still important in the early training stage since the precise evaluation for a given policy is needed to yield the policy improvement.
>
> - Also, Figure 1(b)-(c) should be seen as a whole, rather than individually as they are the results for the same training step. Figure 1.(b) shows that two facts in the early training stage: First, model-errors consistently decay unlike TD-errors. Second, TD-errors with MaPER decays much faster than those without MaPER. Figure 1.(c) shows that **the actual return increases as TD-error decreases with our MaPER**. As a result, Figure 1.(b)-(c), as a whole, show that our method outperforms other methods (see Figure 2 in the manuscript).
>
> *************
>
> **[Q4]** It is also doubtful whether a representation can describe the dynamics imply it can predict Q values.
>
> **[A4]** Our motivation came from the definition of Q-value equation (see Eq. (3.13) in [1]). If the environment models are fully known, Q-values can be calculated exactly (in theory). Due to this fact, we assumed that learning to predict the models leads to the improved representation for predicting Q-values.  Besides, the **prediction of environment models is relatively easier** than that of Q-values because the former is the **one-step** estimation while the latter requires **multi-step** estimation for a long horizon. Therefore, there could be a curriculum learning effect, where the representation learned for model estimation helps learn the Q-value more accurately.
>
> [1] Sutton, Richard S., and Andrew G. Barto. Reinforcement learning: An introduction. MIT press, 2018.
>
> ************
>
> **[Q5]** I am doubting it has some exploration effect.
>
> **[A5]** Indeed, MaPER seems to stimulate exploration while training. For instance, Figure C.6 shows experiences while training after 50,000 environment steps on Sparse-Pendulum, which we designed. The agent with MaPER controls the rod to various directions (upward, downward, and horizontal) for receiving rewards, even if the agent never receives a reward before. However, the agent with the vanilla PER only controls the rod to the downward direction. In other words, MaPER’s coverage of Sparse-Pendulum’s observable space is much higher than PER’s. We thought that learning and computing scores with additional auxiliary learnable features stimulates agents’ exploration.
>
> *************
>
> **[Q6]** The model error’s behavior seems to be domain-dependent.
>
> **[A6]** We agree, but our method is still domain-independent, since irrespectively of domains, predicting rewards and dynamics is relatively easier than predicting Q-values, as the former and the latter involve one-step and multi-step estimations, respectively. In the case of high dimensional action and state spaces, the prediction of rewards and dynamics is more difficult as you mentioned. However, we believe that the prediction of Q-values is even harder in that case.

---

> > ### Comment · Reviewer_noEm · 2021-11-22
> > **Thanks for the response**
> >
> > Thanks for the detailed response. I have not decided yet if I want to improve the rating or not. I suggest the authors make the following improvements, no matter the paper will get accepted or rejected by this conference.
> >
> > Whether the model is easier (than the value function) to learn or not is problem-dependent. You don't have to make arguments like "learning model is easier" in the paper because the statement itself lacks clarity (i.e., how to define easiness here), and your experiments do not really support this claim. In fact, even though the model is quite challenging to learn, your method could still be helpful intuitively if you interpret it as a measure of uncertainty.
> >
> > In the ablation study, the version without parameter sharing is worse than the vanilla SAC during the very early learning stage. Note that the only difference between this method and SAC is that the former uses the priorities with model loss. As a result, it seems to tell me that the main benefit of your method comes from the representation (learned with the auxiliary model loss), not the prioritization with model loss. Then the further question is to figure out which part of the model loss (reward or dynamics) affects the early learning behavior the most.
> >
> > As a pure empirical work, it is also important to tell people the limitations of the proposed method, when it works well, and when it cannot work well.
> >
> > "sampling from TD error is far from optimal." It is a strong statement. Your cited references do not really support this statement. It is better to remove it or tune it down. It is unclear so far what the optimal sampling distribution should be.

---

> > > ### Author Response · Authors · 2021-11-23
> > > **Response to the Reviewer noEm's updated comments**
> > >
> > > Thank you for the response including your detailed suggestions to improve our work. We respond to each of your comments below, and have reflected your comments in the revision (revised texts are highlighted in red).
> > >
> > > ********************
> > >
> > > **Some statements are too strong:**
> > >
> > > Following your suggestion, we have **toned down** on some statements as follows:
> > >
> > > * **The third paragraph in Section 1 in the revised manuscript:**
> > >
> > > sampling based only on high TD-errors may be far from being optimal even for training critic networks in some tasks. $\rightarrow$ sampling based only on high TD-errors may be often ineffective or even degrade the sample-efficiency of the RL framework on some tasks, compared to the uniform sampling.
> > >
> > > * **The fourth paragraph in Section 1 in the revised manuscript:**
> > >
> > > The model is expected to be easier to estimate than $Q$-value. Hence, at an early stage of training (when MaCN has poor estimation on $Q$-value), prioritizing samples of high model estimation under MaPER makes MaCN predict the model well. $\rightarrow$ We expect that prioritizing samples additionally with high model estimation errors helps decrease the model errors at the early stage of training, even if MaCN's estimation of the $Q$-value is poor.
> > >
> > > **********************
> > >
> > > **Limitation:**
> > >
> > > We agree that providing the limitation of our method will be helpful for readers, and have added the following paragraph after introducing our method's advantages in Section 5: Discussion and Conclusion, in the revised manuscript:
> > >
> > > "Despite the advantages above, unfortunately, there is no theoretical analysis of under what conditions and domains our method is effective. As a result, in certain complicated environments with high-dimensional state and action spaces for which the model training is extremely difficult, our method may achieve lower performance than that of the model-free baselines. In such environments, MaPER may require an excessive amount of interactions with the environments to train the model estimators. However, in such extreme cases, all-model based methods will be similarly ineffective"
> > >
> > > **********************
> > >
> > > **Representation Learning vs MaPER:**
> > >
> > > As you pointed out, during the very early learning stage, MaPER without parameter sharing uses more time steps to additionally learn model components and may harm the performance, in which case parameter sharing can mitigate the issue. Nevertheless, **MaPER is crucial for obtaining improved performance in the long run**, which is supported by the following table comparing different versions of SAC's performance after 2M time steps.
> > >
> > > \begin{array}{cc}
> > > \hline
> > > \text{SAC performance after 2M time steps} & \text{Mean (Std.)}       \newline \hline
> > > \text{MaPER + MaCN (parameter sharing)} & 2739.87\text{ }(151.95) \newline
> > > \text{MaPER}                   & 2620.89\text{ }(504.59) \newline
> > > \text{MaCN (parameter sharing)}        & 2366.75\text{ }(447.00) \newline
> > > \text{The vanilla}               & 2254.52\text{ }(433.02)
> > > \end{array}
> > >
> > > 1. We can observe that MaPER  improves the **final performances** of SAC with or without parameter sharing.
> > > 2. The difference of **final performances** between SAC + MaPER with and without parameter sharing is relatively marginal (as also reported in Figure 4.(c) in the revised manuscript).
> > > 3. The **final performances** of SAC + MaCN (parameter sharing) with the uniform random sampling is very similar to that of the vanilla SAC (as also reported in Figure 4.(a) in the revised manuscript).
> > >
> > > These results show that using parameter sharing alone without MaPER does not obtain improved final performance, although they help with sample-efficiency in the early straining stage, and the performance improvements mostly comes from the use of MaPER.

---

### Official Review · Reviewer_mk5C · 2021-11-06

**Correctness:** 3
**Technical Novelty And Significance:** 4
**Empirical Novelty And Significance:** 4
**Recommendation:** 8
**Confidence:** 3

**Main Review:**

Writing: In general, the writing is very clear. With that being said, I do have a few questions.
1. In Algorithm 1, The index set $\mathcal{I}$ is never used after initialization.
2. When are the priorities in $\mathcal{P_B}$​ updated?
3. In Eq (10), why are all losses weighted equally?
4. How is MaPER combined with MBPO?

Experiments: The experiments are quite extensive. I do appreciate the experiments of applying MaPER on different algorithms.

I would like to have another ablation to check the effectiveness of MaPER: Can we use MaPER without MaCN? For example, we have two separated networks, one of which predicts Q and another predicts the next state and reward. This method has a larger computational overhead than vanilla PER, but can better decompose MaCN and MaPER.

Does PER in the experiments (Fig 2, Fig 4) use (4) to stimulate exploration? If not, the comparison is not very fair.

Novelty: The method sounds novel. The idea of MaCN is pretty natural so it might exist in some prior work but I'm not sure. Using model/reward prediction error for prioritized experience replay is indeed novel.

**Summary Of The Paper:**

This paper studies the problem of experience replay. It proposes Model-augmented Prioritized Experience Replay (MaPER), a novel experience replay method, based on the intuition that the model is easier to estimate than Q-value. It also proposes a modification to critic network, Model-augmented Critic Network (MaCN), by predicting the reward and dynamics model additionally. Experiments on MaPER show that MaPER can be applied to both discrete action space (DQN) and continuous action space (SAC), both model-free RL algorithms (SAC) and model-based RL algorithms (MBPO), as well as sparse reward tasks, and improve the baseline algorithms a lot.

**Summary Of The Review:**

Given the novelty of the proposed method and extensive experiments, I like this paper and would recommend acceptance.

---

> ### Author Response · Authors · 2021-11-18
> **Response to Reviwer mk5C**
>
> We deeply appreciate your efforts and insightful comments to improve the manuscript. We respond to each of your comments one-by-one in what follows. In the revised manuscript, we have marked the revisions with “green.”
>
> **********
>
> **[Q1]** Writing: In general, the writing is very clear. With that being said, I do have a few questions.
> 1. In Algorithm 1, The index set I is never used after initialization.
> 2. When are the priorities in $P_{B}$ updated?
> 3. In Eq (10), why are all losses weighted equally?
> 4. How is MaPER combined with MBPO?
>
> **[A1]**
> 1. As you pointed out, the index set I is unnecessary, so we have removed it in Algorithm 1 of the revised manuscript.
> 2. $P_{B}$ is updated whenever the critic networks are trained (as like the vanilla PER).
> 3. This is a typo; Adaptively varying coefficients should be considered in Eq. (10) as you mentioned.  We update Eq. (10) in the revised manuscript.
> 4. This is explained in Section B of the supplementary material. To use MaPER in MBPO, we modified the critic networks in SAC, which is the core of MBPO, to MaCN for adding the model-learning components. Notice that MBPO utilizes two buffers to collect true and virtual experiences, respectively. Then MaPER is used to compute both true and virtual experiences’ priority scores, which are updated whenever updating SAC networks.
>
> **********
>
> **[Q2]** Can we use MaPER without MaCN?
>
> **[A2]** Yes, but using MaCN is more effective; we have added Figure 4.(c) in the revised manuscript, which shows learning curves with and without parameter sharing. The results show that even without parameter sharing, calculating priority scores additionally with model-errors is quite meaningful to improve the performance because auxiliary learnable features can be used to sample various meaningful experiences for training compared to the single feature (TD-error).
>
> **********
>
> **[Q3]** Does PER in the experiments (Fig 2, Fig 4) use estimated rewards to stimulate exploration?
>
> **[A3]** PER in Figure 2 and Figure 4 use true rewards and estimated rewards, respectively. Namely, we compared PER and MaPER under estimated rewards in Figure 4.

---

### Author Response · Authors · 2021-11-19
**General Response**

Dear reviewers and AC,

We really appreciate all the reviewers for their constructive comments. We have responded to the common comments as well as individual comments from the reviewers below, and believe that we have successfully responded to all of them. Here we briefly summarize the updates we have made to the revised version of the paper:

* We added empirical comparison about parameter sharing (Figure 4.(c)).
* We inserted wall-clock time analysis (Table 3).
* We inserted holistic performance comparison (Table C.2).
* We analyzed the sensitivity of PER parameters (Figure C.5(a)).
* We analyzed the sensitivity of coefficients in the multi-objective loss (Figure C.5.(b)).
* We compared methods in a noisy environment (Figure C.5.(c)).
* We compared transitions between MaPER vs PER (Figure C.6) for checking state coverage.
* We provided our method’s limitation in Section 5.
* We revised the third and fourth paragraphs in Section 1 to tone down some statements.

In the revised manuscript, we have marked the revisions with “green” (for the first responses) and "red" (for the second responses).

We sincerely believe that these updates may help us better deliver the benefits of the proposed algorithm to the ICLR community.

Thank you very much,

Authors.

---

### Comment · Area_Chair_eA62 · 2021-11-28
**Any final thoughts?**

Dear reviewers,

November 29th is the end of the discussion period. If you have any final thoughts, this is the right time to express them.

Thank you,
Area Chair

---

### Decision · Program_Chairs · 2022-01-20

**Decision:**

Accept (Poster)

**Comment:**

This work proposes a new strategy for prioritized experience replay. It is based on the argument that the TD error itself may not be a good indicator for priority, so we should rely on other factors that are easier and more reliable to learn. The new method is based on two modifications: (1) modifying the critic's objective so that it learns a good model of the environment (reward and transition dynamics) as well, and (2) use the combination of the TD loss and the model loss in order to define the priorities in the ER queue.

The majority of reviewers are positive about this work. They believe the method is novel and the experiments are extensive. The authors improved the paper during the discussion phase, so many of the questions have already been answered.

There are some concerns though, some of them shared by reviewers and some after my own reading of the paper:

One concern is about the justifications for the method, which are based on the heuristic and intuitive arguments, rather than principled approach. Currently it is not clear, at least to me, why adding a model error to the objective is a good idea.

Another concern, which is not shared by the reviewers, is that there is much overlap in the confidence band of figures and confidence intervals of tables. For example, in many of the subfigures of Figure 2 or 4, there is a significant overlap in the shaded areas. Or many of the numbers in Table 1 (with and without MaPER) are within each others' confidence intervals. Are the results statistically significant?

Another comment, again not shared by reviewers, is regarding how the loss functions are defined. Consider the loss in Eq. (1):
- Is there a squared missing? I assume that it is missing. Although it does not matter at this stage, when you add other terms to the loss, it would, i.e, the minimizer of $f(x) + g(x)$ is not necessarily the same as the minimizer of $f^2(x) + g^2(x)$.

- Is the target value based on a fixed parameter (not optimized), or do you actually consider the expected of the TD error, which would be equal to the empirical Bellman error. If the latter, it would be a biased estimate of the Bellman error. And it is not what DQN or the TD method optimizes (that's why Sutton and Barto's textbook calls them pseudo-gradient).

These requires some clarifications.

Another question related to the model: Is it assumed that the model $T_\theta$ is deterministic and predicts a next-state, as opposed to predicting a distribution over them? (cf. equations after (5) )?

All strengths and concerns considered together, I believe this is a good paper overall, and can be accepted at ICLR. Hopefully we get a better understanding of what this method is actually doing in the future research.

I have the following suggestions to the authors:
- Perform statistical significant tests on your results. In some cases, it might be helpful to increase the number of runs from five to a larger number. It may also be more visually clear to provide standard error instead of standard deviation.
- Clarify issues about the definition of the loss function.
- Please consider improving and clarifying your argument of why you method works.
- Please consider the remaining comments by reviewers in order to improve your paper.